# Mutagenesis and structural modeling implicate RME-8 IWN domains as conformational control points

**Anne Norris[1]\*, Collin T. McManus[1], Simon Wang[1], Ruochen Ying[1], Barth D. Grant[1,2]\***

**1** Department of Molecular Biology and Biochemistry Rutgers University, Piscataway, New Jersey, United States of America, **2** Rutgers Center for Lipid Research, New Brunswick, New Jersey, United States of America

\* anorris@dls.rutgers.edu (AN); barthgra@dls.rutgers.edu (BDG)

## Abstract

After endocytosis, transmembrane cargo is differentially sorted into degradative or recycling pathways. This process is facilitated by recruitment into physically distinct degradative or recycling microdomains on the limiting membrane of individual endosomes. Endosomal sorting complexes required for transport (ESCRT) mark the degradative microdomain, while the recycling domain is marked by the retromer complex and associated proteins RME-8 and SNX-1. The separation of endosomal microdomains is also controlled by RME-8 and SNX-1, at least in part via removal of degradative component HRS/HGRS-1 from the recycling microdomain. This activity is likely due to recruitment and activation of chaperone Hsc70 on the endosome by the RME-8 DNAJ domain. To better understand the mechanism of RME-8 function we performed a new phylogenetic analysis of RME-8 and identified new conserved sequence features. In a complementary approach, we performed structure-function analysis that identified the C-terminus as important for microdomain localization and likely substrate binding, while N-terminal sequences beyond the known single N-terminal PH-like domain are important for endosome recruitment. Random mutagenesis identified IWN4, and by analogy IWN3, to be important for the autoinhibitory DNAJ domain binding, with IWN3 playing a critical role in HRS uncoating activity. Combining AlphaFold structural predictions with *in vivo* mutation analysis of RME-8, we propose a model whereby SNX-1 and the IWN domains control the conformation of RME-8 and hence the productive exposure of the DNAJ domain. Furthermore, we propose that the activation of RME-8 is cyclical, with SNX-1 acting as an activator and a target of RME-8 uncoating activity.

## Author summary

The cells of eukaryotic organisms, including animals, plants, and fungi contain several specialized membrane-bound compartments which perform functions essential for life. Trafficking of membrane proteins between these compartments is an active area of study with important implications in human metabolic and neurologic disease. Endosomes are remarkable compartments where transmembrane proteins taken in from the cell's outer

**Data Availability Statement:** All relevant data are within the paper and its Supporting Information files.

**Funding:** BDG is supported by National Institutes of Health, https://www.nih.gov (grant

5R01GM135326) and SW, RY, and CM were supported by Aresty Fellowship for Undergraduate research, https://aresty.rutgers.edu/programs/fellowships. The funders had no role in study design, data collection and analysis, decision to publish, or preparation of the manuscript.

**Competing interests:** The authors have declared that no competing interests exist.

membrane are sorted for degradation or reuse. This sorting is in part achieved by physically distinct protein coats, the degradative and recycling microdomains. Little about how these microdomains segregate from each other is known. In this work, we reveal how recycling protein RME-8, a key player in microdomain segregation, works. We report that RME-8 is likely inhibited on endosomal membranes by homo-oligomerization until it is activated by companion recycling protein SNX-1. This activation allows RME-8 to un-coat both degradative sorting proteins to limit of the microdomains and its own activator SNX-1 as a means of negative feedback regulation. RME-8 and SNX-1 do not act alone, however, but are parts of an expansive recycling machinery, with much exciting work yet to be done to uncover their functions within this broader context.

## Introduction

Endosomes are organelles that play an essential role in protein and lipid sorting in all eukaryotic cells. Over the past 20 years evidence has emerged that microdomains of endosomal limiting membranes are key features of endosomes, with specific coat complexes representing competitive activities of the endosome that define these microdomains. Physical self-associations and/or oligomerization as is seen with HRS/HGRS-1 can partially explain the segregation of microdomains (For review see [1]). Additional cross-regulatory interactions between degradative and recycling microdomains also occur [2–4] but little is known of the molecular mechanisms that separate these coat complexes to maintain efficient endosome function. In our previous work we identified RME-8 as a key protein in this process, acting to keep recycling and degradation in balance.

RME-8 was first identified in our screen for Receptor Mediated Endocytosis (RME) mutants in *C. elegans* [5,6], RME-8 is a conserved endosomal regulator required for cargo sorting [4,7–12]. It functions with Sorting Nexin 1 (SNX-1) and the Retromer complex in endosome to Golgi recycling [4]. Together RME-8 and SNX-1 also negatively regulate the ESCRT complex that mediates degradation on the same endosomes [1,2,4].

Endosome to Golgi sorting of transmembrane cargo occurs during the early to late endosome transition [13–15] (reviewed in [16]). Indeed, the recruitment of Retromer is dependent upon RAB-7, which is recruited to maturing endosomes during this transition [13]. The Retromer associated RME-8/SNX-1 marked recycling microdomain, and the ESCRT-0 marked degradative microdomain, are found adjacent to one another on endosomes during this transition [2]. The formation and separation of such endosomal microdomains are most easily studied in the context of *C. elegans* scavenger cells called coelomocytes, due to their naturally large endosomes that are typically more than 1 micron in diameter [2]. In this system we previously showed that RME-8 is required to preserve separation of recycling and degradative microdomains on sorting endosomes. In particular, RME-8 prevents overassembly of ESCRT-0 that encroaches on and mixes with recycling microdomains when RME-8 is missing [1,2]. This control of microdomains is a unique feature of RME-8 and SNX-1, as mutants lacking retromer components *vps-35* and *snx-3* have normal microdomain separation [2]. Like several other proteins involved in membrane trafficking, rare alleles of RME-8 have been implicated in neurodegenerative disorders such as Parkinson's Disease and Essential Tremor [17,18] (reviewed in [19–21].

RME-8 is a large 260 kDa DNAJ domain protein with an N-terminal PI(3)P lipid binding domain and a C-terminal Retromer associated SNX-1 binding domain. Four conserved IWN repeats, named for their central isoleucine, tryptophan, and asparagine residues, are dispersed throughout the protein, with two on either side of the central DNAJ-domain. These repeats are a defining feature of RME-8, found in all RME-8 homologs, but not other proteins.

The DNAJ domain is a 70 amino acid helical hairpin that recruits and activates the ATPase activity of Hsp70 enzymes. In Eukaryotes the Hsp70/DNAK protein family mediates local melting of three-dimensional protein structures. This activity promotes proper protein folding/refolding, solubilization of protein aggregates, assembly and disassembly of oligomeric structures, and translocation across membranes (reviewed in [22]).

The complexity and number of DNAJ domain proteins has increased tremendously over the course of evolution [23–25], with *C. elegans* having 34 members and Humans having over 50. The 260 kDa RME-8 protein is by far the largest DNAJ domain protein, most of which are small proteins of about 20 kDa. RME-8 is remarkably conserved throughout Eukarya with notable absences in Fungi and Gymnosperms (this study).

While RME-8 represents an important endosomal regulator balancing recycling and degradative activities, how RME-8 itself is regulated has been unclear. We previously noted evidence for a physical interaction between the RME-8 DNAJ domain and sequences C-terminal to the RME-8 DNAJ domain [4]. We posited that the RME-8 C-terminus could occlude the productive DNAJ domain interaction with Hsc70. Another clue to RME-8 regulation is our previous observation that RME-8 and SNX-1 physically interact, with neither appearing to depend upon the other for endosomal recruitment [1,2,4].

In this study we identify new RME-8 regions of interest conserved across phyla. Using structure function analysis, we show that the C-terminus of RME-8 both forms an inhibitory RME-8 C-terminus/ DNAJ domain interaction and controls microdomain segregation *in vivo*. Furthermore, we provide evidence that mutants in the IWN3 domain, but not IWN4, produce an RME-8 molecule hyperactive in uncoating ESCRT-0. Combining structural modeling with RME-8 mutagenesis and *in vivo* microdomain analysis we propose a model whereby SNX-1 disrupts an autoinhibited conformation of RME-8 mediated by its own IWN domains. We posit that this disruption is needed for productive exposure of the DNAJ domain for uncoating of both its own activator, SNX-1, and ESCRT-0 microdomains.

## Results

### RME-8 has ancient origins, high conservation of key domains, and new regions of interest

To better understand RME-8 and its domains we comprehensively analyzed its evolutionary conservation in the 4000 predicted proteomes available at NCBI. We used *C. elegans* RME-8 as a query for a BLAST search, then filtered for >40% query coverage and <1E-3 e-value (see Methods). We found that RME-8 is scattered among Eukarya, as homologs are found in a diverse array of protists, such as the orphan protist lineage *Guillardia theta*, and in both free-living and parasitic protists such as *Entamoeba* and *Dictyostelium*, but absent in *Trichomonas*, *Giardia* and Alveolates (Fig 1A). Interestingly, RME-8 is completely absent from Fungi, but present in the closely related *Choanoflagellates* and Sponges. Additionally, RME-8 is present in Angiosperms but absent in all Gymnosperms save for the Chinese Yew. It is important to note that this type of analysis is a snapshot in time and will change as a more diverse array of organisms are sequenced and our understanding of their evolutionary relationships change. However, this pattern does suggest an ancient origin for RME-8.

Given the broad array of sequences compared, our analysis narrows and illuminates key domains in RME-8. Not surprisingly the DNAJ-domain is a highly conserved feature, as DNAJ domains are ancient and widely conserved in all kingdoms of life (Figs 1B,1C and S1). Despite being well conserved in animals, both the extreme N-terminal lipid binding domain, as defined by [10], and the IWN1 repeat, display low sequence conservation across Eukarya. The IWN2, IWN3 and IWN4 repeats, however, display strong conservation throughout

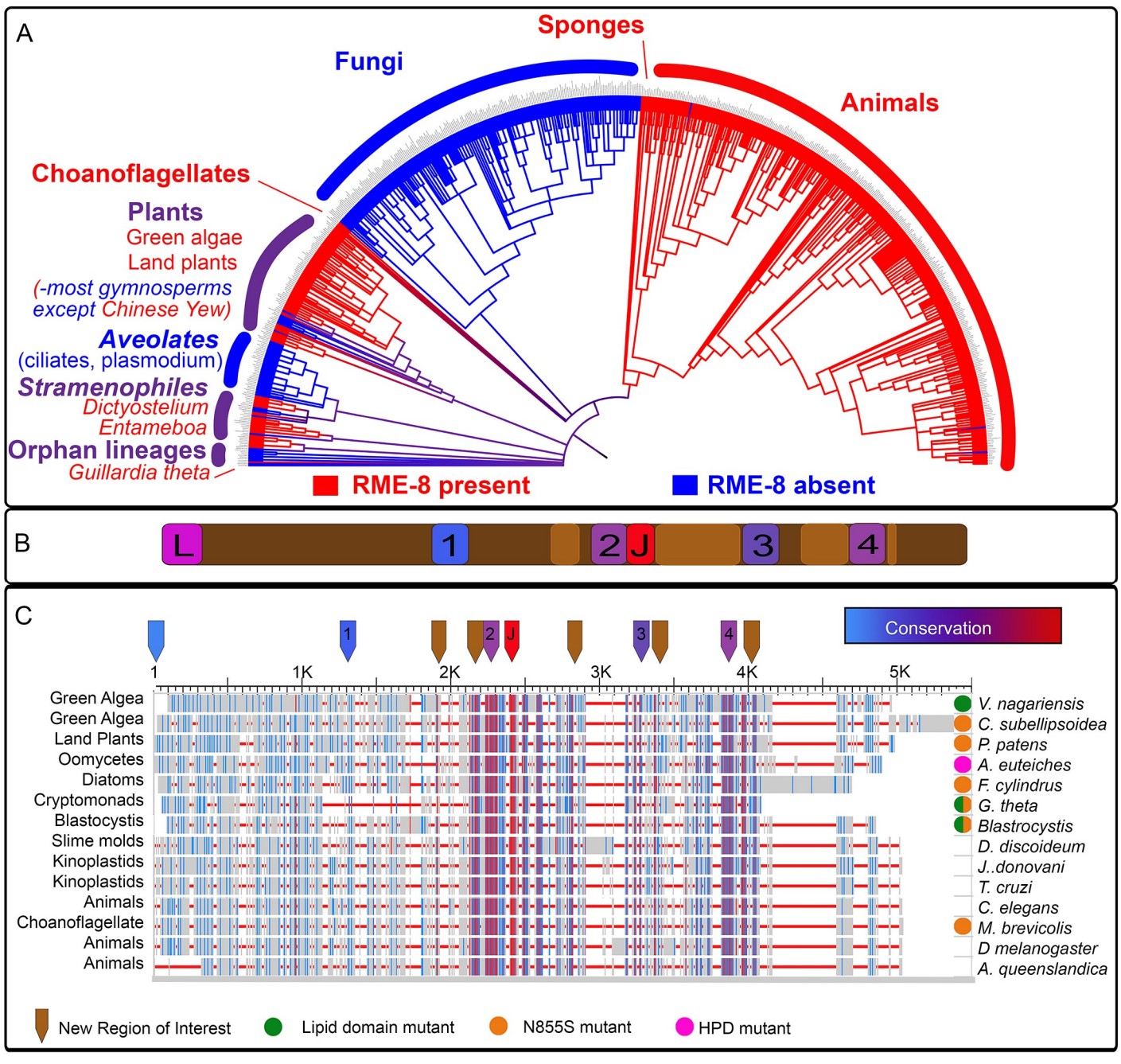

**Fig 1. RME-8 homologs are found dispersed throughout Eukarya.** (A) A phylogram of Eukarya generated in iTOL, (see methods). Branches containing an RME-8 homolog are indicated in red, nodes containing branches with RME-8 homologs present and absent are colored in purple, and branches or nodes containing no RME-8 homologs are colored in blue. (B) A schematic of RME-8 with key domains indicated with new regions of interest (from the MSA) indicated in light brown and the colors of the IWN domains are in accordance with their conservation as seen n C. Lipid binding domain (L), IWN domains (1–4) and DNAJ (J) are indicated. (C) An alignment of a sampling of RME-8 homologs from dispersed branches in Eukarya generated by the NCBI Multiple Sequence Alignment (MSA) viewer. The names of the phylogenetic categories are indicated to the left, and species names are indicated to the right of the MSA. Colored dots indicating mutations in key domains are indicated to the left of the species name. A pink dot indicates that key residues in the DNAJ domain are mutated. A green dot indicates the key W20 (human) residue in the originally proposed lipid binding domain is absent, and an orange dot indicates the Parkinson's associated residue N855 (human) is not conserved. In the MSA, red horizontal lines represent gaps, grey bars represent minimal homology, blue bars represent low homology and red indicates high homology. Arrows above the alignment indicate key areas of interest, known regions of interest are colored based on their homology, new regions of interest are indicated by light brown arrows. The colors of the IWN domain arrows are in accordance with their conservation.

Eukarya (Fig 1C). The regions from the RME-8 N-terminus to just before IWN2 are much less conserved than the remaining C-terminal sequences (Fig 1C). Moreover, the Parkinson's associated residue N855 is only conserved in animals (Fig 1C).

Our analysis also reveals new areas of interest outside of the defined functional domains. Specifically, residues just adjacent to IWN2 are highly conserved, as are sequences between the DNAJ-domain and IWN3. Additionally, the region between IWN3 and IWN4 displays strong conservation (See Figs 1C and S1). In the current study we set out to better understand the function of the IWN domains of RME-8 with respect to their role in controlling endosomal microdomains. Furthermore, we propose potential functions for the newly highlighted regions and IWN 2–4 domains.

## The RME-8 IWN regions are important for membrane localization and microdomain positioning

The position of endosomes is stereotyped in the six specialized *C. elegans* scavenger cells called coelomocytes. RAB-5 marked early endosomes are located toward the periphery of the radially symmetrical disc shaped coelomocyte (Fig 2B, illustrated in Fig 2A). RAB-7 marked late endosomes and lysosomes localize more toward the interior (Fig 2C, illustrated in Fig 2A) [6,26]. The Golgi forms dispersed ministacks typical of invertebrate cells, and the nucleus is positioned centrally (See Fig 2A). RME-8 localizes to recycling microdomains of the early endosomes, most easily observed in coelomocytes because their endosomes are naturally quite large (1–5 micron diameter) [1,2]. We find that the bulk of RME-8 microdomains on peripheral endosomes are oriented toward the plasma membrane (Fig 2D).

To further study the role of the IWN regions for RME-8 function *in vivo*, we created a series of deletion transgenes that express tagged RME-8 missing certain IWNs and their surrounding sequences (Illustrated in Fig 2J). We measured how the RME-8 truncations compared to wild-type in their ability to localize to discrete microdomains on endosomes of the coelomocyte. RME-8 retaining the established lipid binding domain [10] but missing other N-terminal regions (Δ aa100-1321; referred to as ΔIWN1+2r) is less membrane localized than wild-type RME-8, with more diffuse labeling in the cytoplasm (Fig 2G and 2J, quantified in Fig 2L). This result suggests that N-terminal sequences outside of the previously defined lipid binding domain contribute to endosomal recruitment (see below). Interestingly, the remaining endosomal localization that did occur in this deletion mutant still segregated into peripherally facing microdomains like wild-type RME-8 (Fig 2G, quantified in Fig 2K, Illustrated in Fig 2M)). Taken together these results suggest that the N-terminal amino acids 100–1321 (See Fig 1B) play a role in membrane association, but do not control RME-8 microdomain localization.

In stark contrast, RME-8 lacking the C-terminal IWN3 and/or IWN4 regions; referred to as ΔIWN3+4r (Δ1650–2279), ΔIWN3r(Δ1389–1950), and ΔIWN4r (Δ1950–2279) displayed both more intense endosome localization, and broader spread on the endosomal limiting membrane (Fig 2E, 2F and 2H, quantified in Fig 2K–2L, Illustrated in Fig 2M). Moreover, the physically distinct recycling and degradative microdomains become mixed in the RME-8 IWN3r and/or IWN4r deletion strains (Figs 3C and S3G–S3K", illustrated in Fig 3N, quantified in Figs 3H and S3L). While the localization of RME-8 C-terminal truncation mutant protein spreads around the endosome, its colocalization with tagged SNX-1 is dramatically reduced as SNX-1 localization appears largely unchanged. (Figs 3D, 3F and S3A–S3E", illustrated in Fig 3I, quantified in Figs 3G and S3F).

Taken together, these results indicate that the RME-8 sequences N-terminal to the DNAJ domain, and sequences C-terminal to the DNAJ domain, have distinct functions. N-terminal sequences well beyond the previously proposed lipid binding domain contribute to membrane association, but not microdomain segregation. The microdomain segregation function

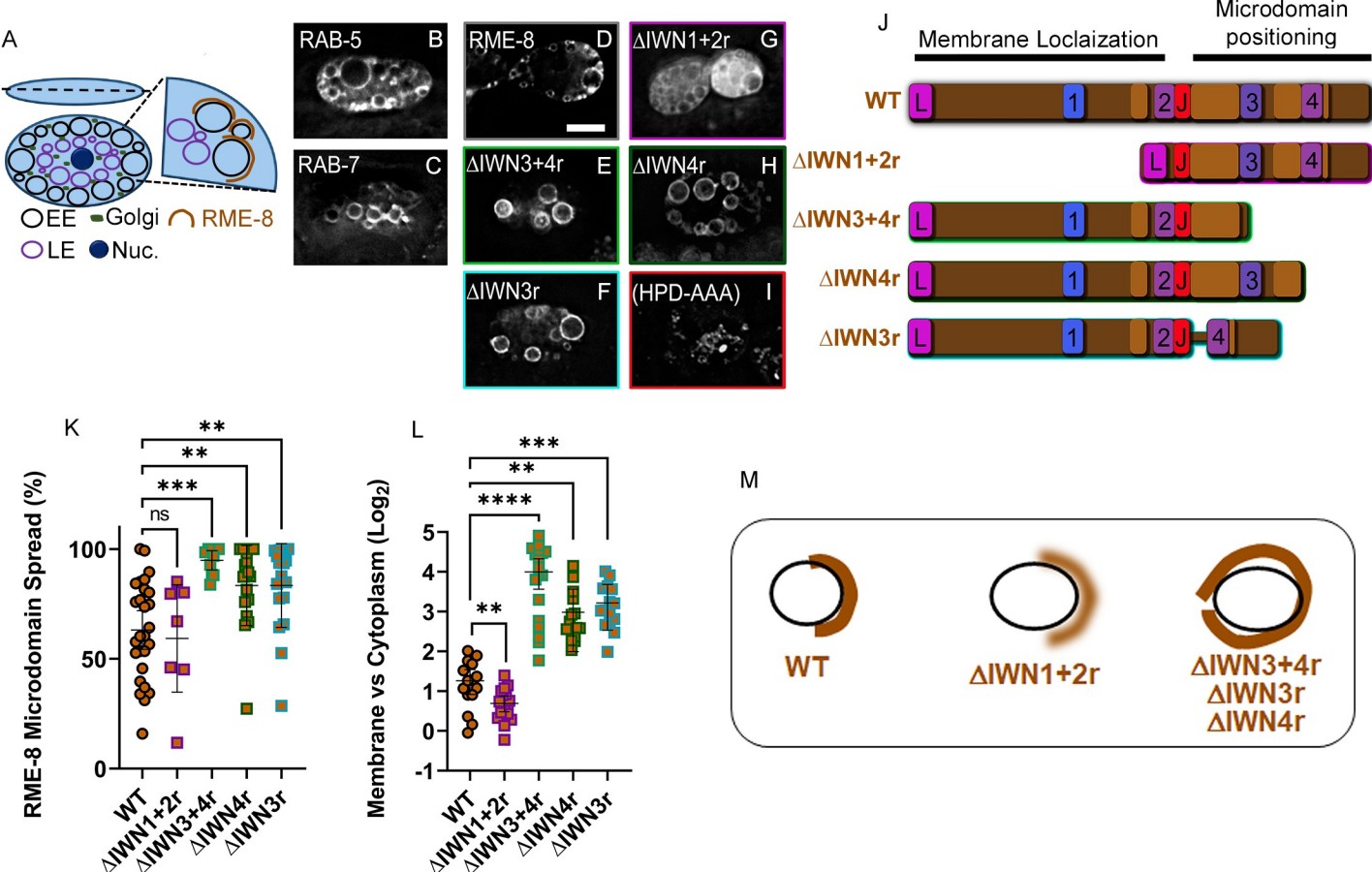

**Fig 2. The RME-8 N-terminus controls membrane association, while the C-terminus controls microdomain positioning, and microdomain spread.** (A) Early endosomes (EE), Late endosomes (LE), Golgi, and the Nucleus occupy stereotypical positions in the coelomocyte cell. (B-C) Micrograph of tagRFP::RAB-5 (B) and tagRFP::RAB-7 (C) expressed in coelomocytes. (D-I) Micrograph of pCUP-4::GFP::RME-8 *wild-type* and mutants expressed in wild-type animals. Micrograph of pCUP-4:: GFP::RME-8 N-terminal (G) C-terminal truncations (E, F, and H), and a DNAJ domain mutant where the catalytic triad HPD residues are substitute with alanines (I). (J) To scale illustration of wild-type, N-terminal and C-terminal truncations of RME-8. (K) Ratio of the GFP::RME-8 intensity in the cytoplasm versus on the membrane in the indicated GFP::RME-8 *wild-type* and truncations indicated in J. Given that a ratio is plotted, we used a Log2 scale. (L) Quantification of the percentage of an endosome covered by the indicated GFP-RME-8 *wild-type* and truncation mutants. (M) Illustration of the alterations of the RME-8 microdomain that occur in the truncations illustrated in J. All micrographs are deconvolved widefield images (see methods). In K-L each data point is an individual worm, error bars indicate Mean with 95% CI, ANOVA statistical analysis unless ratios are plotted in which case student t-test are used, done in Graphpad Prism with p<0.01 = **, p<0.001 = ***, p<0.0001 = ****. Scale bars are 5 microns in whole coelomocyte images.

requires both the IWN3 and IWN4 regions, as loss of either leads to RME-8 spreading around the endosome and microdomain mixing (Figs 2D–2H, 3A–3F and S3, quantified in Figs 3G, 3H and 2L, and Illustrated in Figs 2M and 3N). Moreover, given that deletion of IWN3 or IWN4 regions leads to increased membrane localization, as measured by membrane to cytoplasm ratio (Fig 2K), IWN3 and IWN4 regions may contribute to removal of RME-8 from the endosome as it cycles on and off the membrane.

## The RME-8 IWN3 and IWN4 regions are required for RME-8 to uncoat ESCRT-0

A significant role for RME-8 in directing cargo sorting is to limit the growth of the degradative microdomain. Moreover RME-8 prevents the degradative machinery from entering the

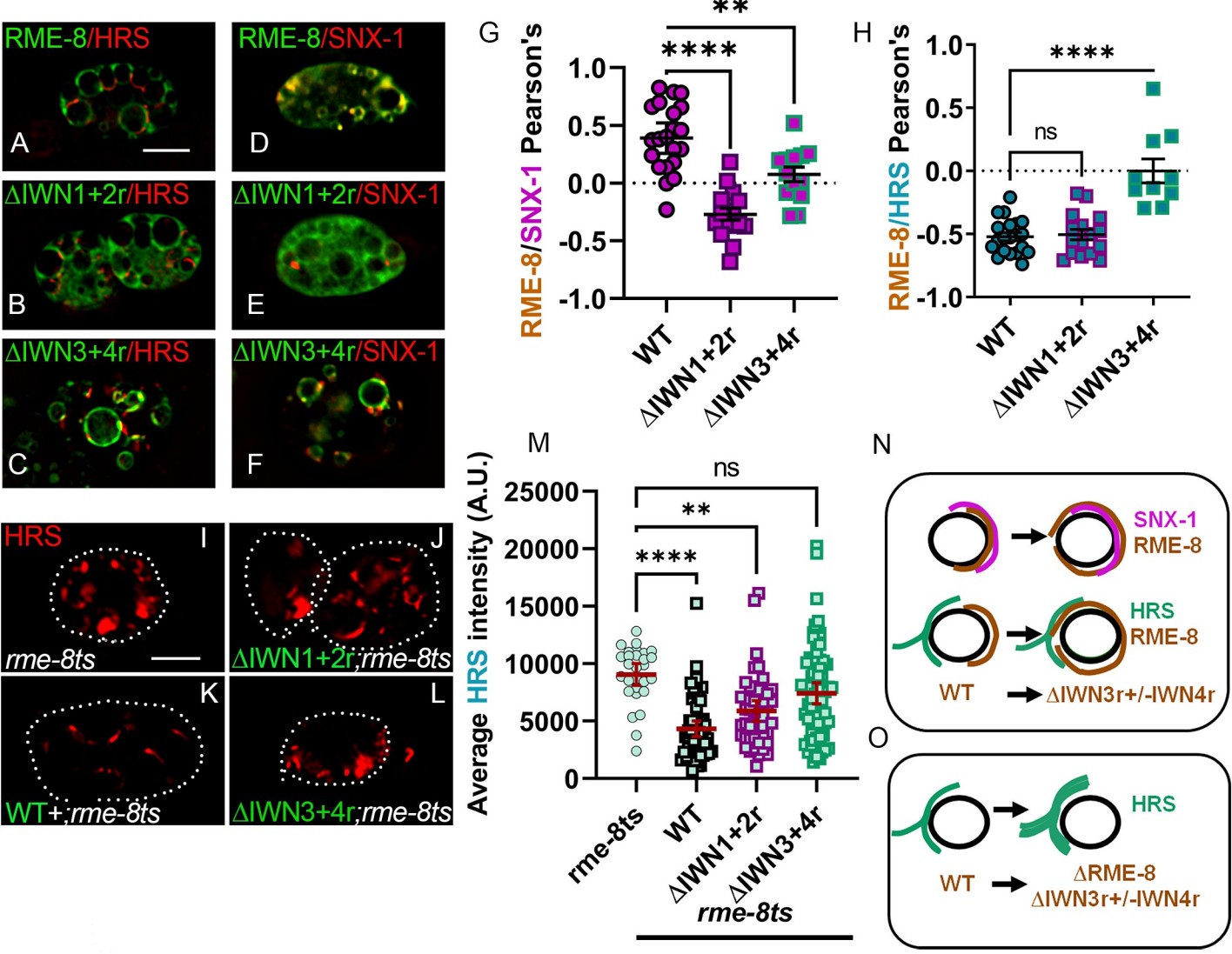

**Fig 3. RME-8 C-terminal IWN3+4 regions mediate its recycling versus degradative domain segregation.** Both N-terminal IWN1+2 and C-terminal IWN3+4 regions are important for HRS (HGRS-1) uncoating activity. (A-C) Micrograph of pCUP-4::GFP::RME-8 *wild-type*, N-terminal, and C-terminal truncations co-expressed with tagRFP::HRS (HGRS-1) in coelomocytes of *wild-type* animals. (D-F) Micrograph of pCUP-4::GFP::RME-8 *wild-type*, N-terminal, and C-terminal truncations co-expressed with tagRFP::SNX-1 in coelomocytes of *wild-type* animals. (G) Quantification of colocalization of pCUP-4::GFP::RME-8 *wild-type* and truncations indicated in Fig 2 with tagRFP::SNX-1 in coelomocytes of *wild-type* animals. (H) Quantification of colocalization of GFP::RME-8 *wild-type* and truncations with tagRFP::HRS(HGRS-1) in *wild-type* animals. (I-L) Micrograph of TagRFP::HRS(HGRS-1) in *rme-8ts* animals expressing N-terminal IWN1+2 and C-terminal IWN3+/-4 truncations (M) Quantification of tagRFP-HRS(HGRS-1) average intensity in *rme-8ts* animals expressing pCUP-4::GFP::RME-8+, N-terminal, or C-terminal truncations. (N) Illustration of the microdomain segregation that is disrupted when IWN3 and IWN4 regions are deleted in RME-8. (O) Illustration of the accumulation of HRS (HGRS-1) that occurs in the absence of RME-8, and N-terminal IWN1+2 and C-terminal IWN3+/-4 regions. Each data point is an individual worm, error bars indicate Mean with 95% CI, ANOVA statistical analysis done in Prism with p<0.5 = *, p<0.01 = **, p<0.001 = ***, p<0.0001 = ****. Scale bars are 5 microns in whole coelomocyte images.

recycling microdomain, consequently limiting degradation of endocytosed cargo that should recycle [2]. In the absence of RME-8, ESCRT-0 component HRS/HGRS-1 over accumulates on the endosomes, and mixes with the recycling domain [2].

We found that RME-8 mutants lacking IWN3 and IWN4 regions fail to rescue the HRS/HGRS-1 overaccumulation phenotype of *rme-8(b1023ts)* mutants (Fig 3I–3L, quantified

Fig 3M, illustrated in Fig 3O). This inability to uncoat HRS/HGRS-1 occurs despite a marked increase in RME-8(ΔIWN3+ΔIWN4) spatial overlap with HRS/HGRS-1 (Fig 3C, quantified in Fig 3H, illustrated in Fig 3N). RME-8 N-terminal truncation (ΔIWN1+2r), did however retain a very weak ability to uncoat HRS (Fig 3J, Quantified in Fig 3M). We note that none of the truncation mutants of RME-8 appear fully functional, as they all fail to localize properly (S2A–S2E Fig) or rescue the smaller coelomocyte size (quantified in S2F Fig) phenotype that occurs upon loss of RME-8.

## The RME-8 DNAJ domain inhibits the SNX-1/RME-8 interaction

Our previous work indicated that RME-8 sequences C-terminal to the DNAJ domain interact with both the RME-8 DNAJ domain and to the helical BAR domain of SNX-1 [4]. We postulated that SNX-1 binding to RME-8 C-terminal sequences may compete with the RME-8 C-terminal sequence binding to the RME-8 DNAJ domain, contributing to DNAJ domain regulation (See Fig 4I). Consistent with such a competition, we found that the presence of the DNAJ domain decreased the RME-8/SNX-1 interaction as assayed by yeast two-hybrid (Figs 4A and S4B). This result held true using either a minimal region that only includes the IWN3 region, or the optimal region that includes both IWN3 and IWN4 regions (Figs 4A and S4B). Taken together these results support the hypothesis that the SNX-1 domain competes with the RME-8 C-term/DNAJ interaction and is consistent with an *in vivo* role for SNX-1 in activating RME-8 by increasing the availability of the DNAJ domain to interact with substrates.

## Charge reversals in IWN4 and IWN3 alter the RME-8 C-terminus/SNX-1 and RME-8 C-terminus/DNAJ interactions

To further understand this potential competition, we sought to identify key residues in RME-8 that mediate the DNAJ versus SNX-1 interactions. To this end, we used error prone PCR to create a library of random mutations in RME-8 C-terminus including the DNAJ domain (1322–2279) and screened for those mutations that imparted improved interaction with SNX-1 in yeast 2-hybrid. The RME-8 C-terminal fragment containing the DNAJ-domain paired with SNX-1 grows on the less stringent assay media SC-HIS (S4 Fig), indicating an interaction, but, importantly, does not grow on the more stringent assay media SC-URA (Fig 4B). This difference enabled selection for putative improved RME-8/SNX-1 binding mutants on SC-URA. Given that in PCR mutagenesis the most common errors are premature stop codons and frame shifts, selecting for an improved SNX-1 interaction enriches for informative full-length RME-8 point mutants.

This screen yielded two mutants (E1962K and N1966K), both of which altered residues in the IWN4 domain of RME-8. Suggesting that charge may play a role, both mutations replaced an acidic residue (E1962) or a polar residue (N1966) with the basic residue lysine (Fig 4B and 4C). Likewise, further tests showed that replacing E1962 with arginine also led to growth of the RME-8 bait with SNX-1 BAR domain prey on the SC-URA stringent assay media (Fig 4B). This result suggests that the charge of the IWN4 region is important. Importantly, a separate pulldown assay showed that the IWN4 E1962K mutation significantly reduced binding between the DNAJ domain and the IWN3-IWN4 containing fragment of RME-8, an unselected effect (Fig 4H). This result is consistent with our hypothesis that IWN4 mutations are improving SNX-1 binding by reducing a competing binding reaction with the RME-8 DNAJ domain (Illustrated in Fig 4I).

Given the IWN3 central positioning in the minimal SNX-1 binding domain, as well as its similarity to IWN4, we also tested similar charge reversals of acidic residues in IWN3. We found that introduction of a D1657K mutation in IWN3 also reduced interaction between the

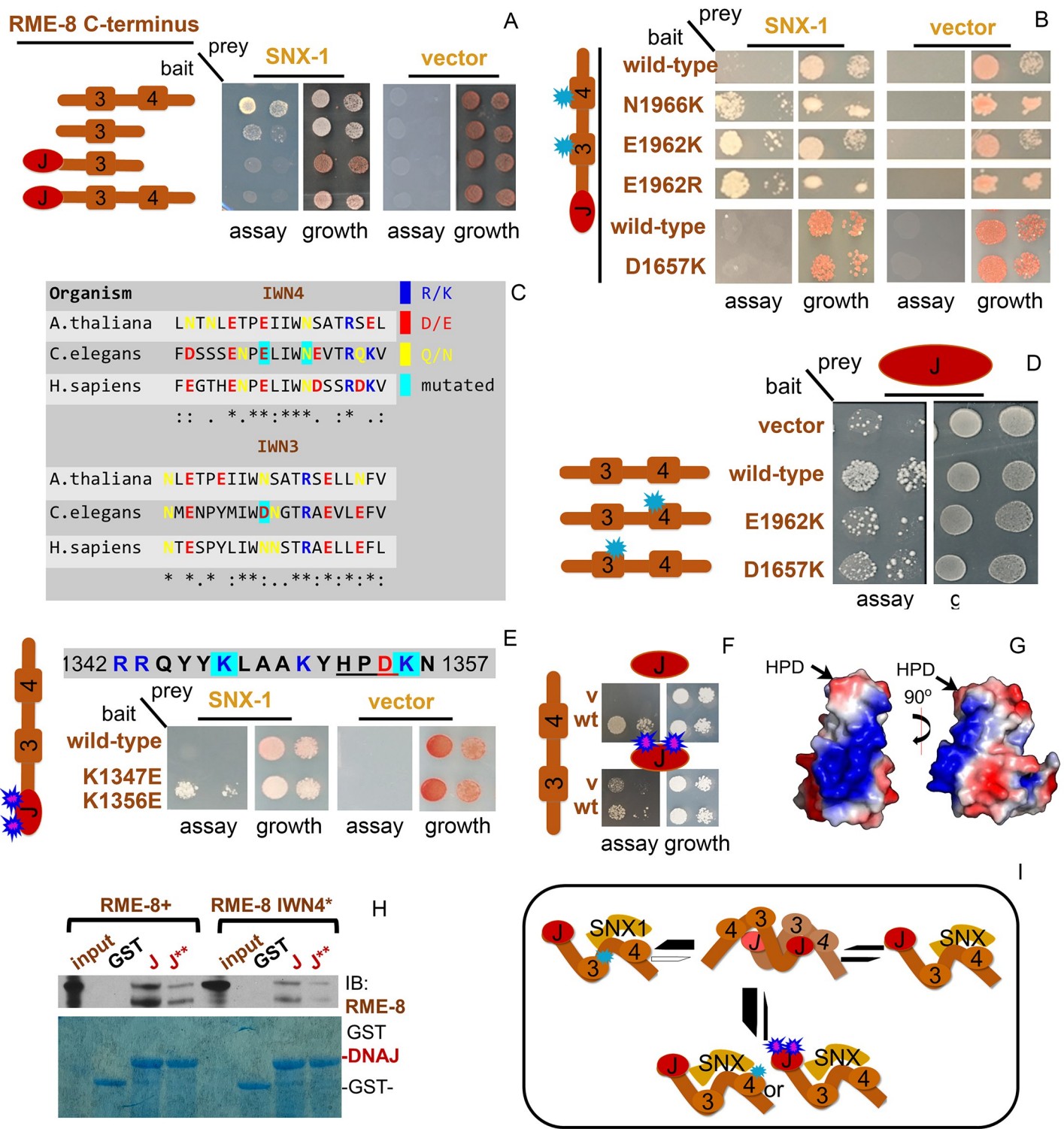

**Fig 4. The RME-8 DNAJ, IWN4, and IWN3 domains mediate SNX-1 and RME-8 self-interactions.** (A) SNX-1 (aa221-472) expressed in pDEST22 tested for interaction with empty vector or fragments of the RME-8 C-terminus expressed in pDEST32 using JDY27 containing URA3, ADE2, and HIS3 reporter genes. 5ul of suspended yeast at 1, and 0. 1 OD's s were spotted on SC-LEU-TRP growth or SC-LEU-TRP-URA assay media. A schematic representation of the RME-8 fragments tested are illustrated to the left of the yeast two hybrid assay. (B) PCR mutagenesis of the RME-8 (DNAJ-end) fragment yielded two mutants in IWN4. Both mutations were lysine substitutions at Glutamic Acid 1962 and Asparagine 1966, that enabled an RME-8(C-term)/SNX-1(BAR) growth on the stringent assay media SC-LEU-TRP-URA. An arginine substitution at position 1962 also enabled growth on the stringent assay media SC-LEU-URA. An analogous IWN4 mutation as was found to alter SNX-1 interaction was

introduced into IWN3. A Lysine substituted for Aspartic acid displayed the same lack of growth on the stringent assay media as *wild-type*. A schematic of the RME-8 fragment is illustrated to the left of the assay. 5ul of suspended yeast at 0.5 and 0.05 OD's s were spotted on SC-LEU-TRP growth or SC-LEU-TRP (-URA or +20mM 3AT) assay media. (C) alignment of IWN4s and IWN3s from *C. elegans*, *H. sapiens* and *A. thaliana*. RME-8 E1962 and N1966 of IWN4 identified in the screen are highlighted with cerulean. An analogous residue D1657 of IWN3 is also highlighted in cerulean. Basic residues are indicated in blue, acidic residues in red, and N/Q polar residues are indicated in yellow (D) Fragments of RME-8 C-terminus (no DNAJ) in the Duplex LexA 2-micron yeast two hybrid system were tested for their ability to interact with the DNAJ domain and the IWN3-IWN4 containing domain. 5ul of suspended yeast at 2, 0.2, and 0.02 OD's s were spotted on SC-HIS-URA-TRP growth or SC GAL-HIS-TRP-LEU SC assay media. IWN3 and IWN4 mutants of the C-terminus displayed diminished interaction compared to wild-type. The isolated IWN4 fragment displayed similar growth to full length. A schematic of the mutants and fragments are illustrated to the left of the assay. (E) Lysines and Arginines of helix II were targeted for doped oligo mutagenesis and selected for increased interaction with SNX-1. The two lysines identified by the screen are highlighted in cerulean. Basic residues are indicated in blue and acidic residues in red. The underline indicates the catalytic triad. The aa1322-2279 fragment with lysines at position 1347 and 1356 substituted with glutamic acid was sufficient to show growth on assay media when combined with SNX-1 BAR domain. A schematic of the Fragment mutated represented to the left of the assay. 5ul of suspended yeast at 1, and 0.1 OD's were spotted on SC-LEU-TRP growth or SC-LEU-TRP-URA assay media. (F) The aa1322-1388 (DNAJ) fragment with lysines at position 1347 and 1356 substituted with glutamic acid disrupted the RME-8/DNAJ interaction. V = vector and wt = wild-type. A schematic of the Fragment mutated represented to the left of the assay. 5ul of suspended yeast at 1, and 0.1 OD's were spotted on SC-LEU-TRP growth or SC-LEU-TRP-URA assay media. (H) Fragments of HA-tagged RME-8 C-terminus (no DNAJ) wild-type and IWN4* expressed in an *in vitro* TnT system were incubated with DNAJ-GST or DNAJ*(K1347E/K1356E)-GST Sepharose beads. The bound fraction was run on an SDS-PAGE gel and probed with an HA-antibody. The incubation reaction was also run on a separate gel and stained with Coomassie Blue to indicate the amount of GST-tagged DNAJ domain. DNAJ** disrupted the RME-8/DNAJ interaction. (I) An illustration of our hypothesis that mutations in DNAJ, IWN4, and IWN3 disrupt RME-8 self-interaction. (G) Three views of an electrostatic representation of the RME-8 DNAJ domain threaded onto PDB file 2OCH. Blue indicates basic, and red indicates acidic surfaces. The HPD catalytic triad of DNAJ domains is indicated by arrows. The center image displays the basic nature of helix II of the DNAJ domain.

DNAJ domain and the IWN3-IWN4 containing domain (Fig 4D). However, unlike with IWN4, D1657K mutation in RME-8 IWN3 did not measurably increase the RME-8/SNX-1 interaction in our assay (Fig 4B). Taken together these data support the idea that the acidic residues of IWN4 and IWN3 play different roles for SNX-1 binding but are both important for a DNAJ domain/RME-8 C-terminus self-interaction.

## Charge reversals near the DNAJ domain active site also alter the RME-8/SNX-1 interaction

If the acidic residues of IWN4 and IWN3 mediate an electrostatic interaction with the DNAJ domain, we would expect to find complementary basic residues in the DNAJ-domain that would have a similar role in informing the SNX-1/RME-8 interaction. Indeed, several lysine and arginine residues on helix II of all DNAJ domains create a surface exposed basic patch (shown in blue) adjacent to the catalytic HPD residues that reside between helix II and III (Fig 4G [23,27–29]). This basic patch has been implicated in Hsc70 binding to DNAJ domains, and thus any domain interactions with this patch could regulate Hsc70 activation [28,30].

With an analogous strategy to that which identified the IWN4 mutants, we targeted the lysines and arginines (Fig 4E) of helix II and III in the predicted RME-8 DNAJ-domain basic patch (Fig 4G). We used doped oligo directed mutagenesis to screen for an improved RME-8/SNX-1 interaction (See methods). Transformants were selected on SC-URA and sequenced. The arginine residues of the DNAJ-domain were screened in a separate but similar fashion.

Our screen identified a K1347E/K1356E double mutant that produced robust growth on SC-URA (Fig 4E). Analogous residues in the *E. coli* DnaJ protein were found to both contribute to DNAK(Hsc70) binding and to be important for the DNAJ *in vivo* activity [28,30]. After a week of growth on SC-URA the R1342E/R1343E double mutant displayed some growth (S4 Fig), also suggesting that the basic nature of the DNAJ domain is important.

If the improved SNX-1 binding of these DNAJ-domain mutants is due to a disruption of the RME-8 C-terminus/DNAJ-domain self-interaction, we would expect these mutants to display decreased ability to bind to the RME-8 C-terminus. To this end we performed GST pull-down assays and yeast two-hybrid assays in which wild-type or mutant DNAJ domain was used as bait with RME-8(1388–2279) prey. We found that the charge reversals in the DNAJ domain weakened its interaction with the RME-8 C-terminus (Fig 4F and 4H). These results

suggest that K1347E and K1356E depress the RME-8 self-interaction in favor of an RME-8/SNX-1 interaction (See Fig 4I for illustration).

## RME-8 D1657K (IWN3*), but not IWN4*, is hyperactive for uncoating HRS/HGRS-1

A key function of RME-8 and SNX-1 is to limit the assembly of the ESCRT-0 microdomain [2]. Hence, we tested the functionality of our newly identified IWN3 and IWN4 altered alleles of RME-8 *in vivo*. Importantly, we found a significant reduction of HRS/HGRS-1 on endosomes in transgenic animals expressing the RME-8(IWN3*) D1657K mutant, but not in IWN4*(Fig 5A–5C", quantified in Fig 5E). Since this is the opposite effect of *rme-8* loss-of-function, these results indicate that RME-8(IWN3*) is hyperactive.

This hyperactivity can likely be attributed to reduced inhibition of the DNAJ domain when IWN3 is mutated. Importantly, RME-8(IWN3*) also shows increased overlap with the normally physically distinct ESCRT-0 microdomain, consistent with more HRS/HGRS-1 engagement and disassembly (Fig 5A–5C", quantified in Fig 5D, illustrated in Fig 5F). Conversely, *in vivo* we observe SNX-1 colocalization with RME-8 is dramatically lower for RME-8(IWN3*) than with either wild-type RME-8 or IWN4* (Fig 6J–6L", quantified in Fig 6M, illustrated in Fig 6N). Our results for IWN4 mutants, that display an increase in SNX-1 binding, were different. Indeed, none of these altered HRS/HGRS-1 accumulation (Figs 5C" and S5A–S5D, quantified in Figs 5E and S5E). We tested CRISPR mediated endogenous alterations E1962K, and a triple lysine substitution at E1962, E1967, and N1966 (S5 Fig). We also tested the IWN4* transgene E1962, E1966, E1967 (Fig 5C"). Taken together these results suggest that RME-8 may in part act independently of SNX-1 for degradative domain uncoating activity.

## RME-8(+) can regulate RME-8(IWN3*)

Interestingly, the hyperactivity of RME-8(IWN3*) in reducing HGRS-1/Hrs levels on endosomes was blocked by endogenous wild-type RME-8 (Figs S5F and S5G, quantified in S5H Fig), since we only observed this hyperactivity of RME-8(IWN3*) when endogenous RME-8 was removed by temperature shift of *(b1023ts)* mutants (Fig 5A" and 5B", quantified in Fig 5E). As mentioned above, the microdomain positioning of the hyperactive RME-8(IWN3*) shows more overlap with HGRS-1/Hrs (Fig 5A-5C, quantified in Fig 5D, illustrated in Fig 5F), and is shifted internally, away from the plasma membrane (Fig 6C, quantified in Fig 6H, illustrated in Fig 6I). This effect could be diminished by overexpression of wild-type RME-8 (Fig 6D–6E" quantified in Fig 6H, illustrated in Fig 6N). The ability of wild-type RME-8 to inhibit the increased uncoating activity of RME-8(IWN3*), and to affect the localization of RME-8(IWN3*), suggests that RME-8 may homo-oligomerize, with wild-type RME-8 able to bind and inhibit RME-8(IWN3*).

## RME-8 endosomal positioning is also altered by expression level and SNX-1

RME-8(+) displays a very peripheral and clumpy microdomain localization in *snx-1(0)* mutants, the opposite of the microdomain localization of RME-8(IWN3*) (Fig 6B and 6C, quantified in Fig 6H illustrated in Fig 6I). These differences suggest that while at some point in its activation cycle RME-8 may act independently of SNX-1, SNX-1 still strongly influences RME-8 localization and activation. Indeed, the increased peripheral localization of RME-8(+) in *snx-1(0)* mutants is similar to higher levels of RME-8(+) overexpression (Fig 6D–6D" quantified in Fig 6H). This result supports the idea of a competition between SNX-1 binding and RME-8 self-binding. These results also appear more compatible with RME-8

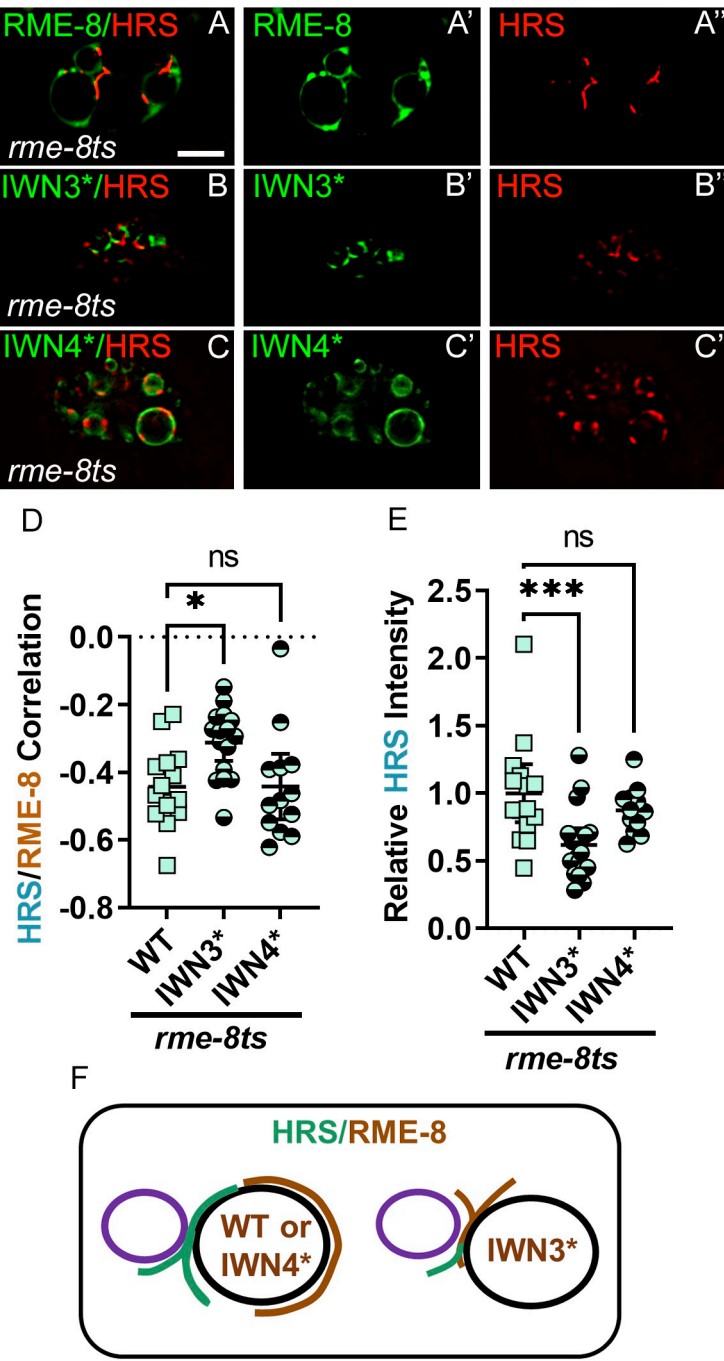

**Fig 5. IWN3\* mutant alters both RME-8/HRS colocalization and HRS accumulation.** (A-A") Micrograph of pCUP-4::GFP::RME-8 co-expressed with tagRFP::HRS(HGRS-1) in coelomocytes of *rme-8ts* animals. Single channels and the merge are displayed. (B-B") Micrograph of pCUP-4::GFP::RME-8 D1657K mutation in IWN3 (IWN3\*) co-expressed with tagRFP::HRS(HGRS-1) in coelomocytes of *rme-8ts* animals. Single channels and the merge are displayed. (C-C") Micrograph of pSNX-1::tagRFP::HRS(HGRS-1) co-expressed with RME-8 IWN4 triple lysine substitution at positions 1962,1966, and 1967 (IWN4\*) in *rme-8ts* mutant animals. Single channels and the merge are displayed (D) Quantification of colocalization of HRS(HGRS-1) with RME-8 WT, IWN3\* and IWN4\*. (E) Quantification of tagRFP::HRS/HGRS-1 average intensity *rme-8ts* mutant animals expressing RME-8 WT, IWN3\* and IWN4\*. (F) Illustration of the interior shift and increased HRS overlap of the RME-8 microdomain that occurs upon the introduction of D1657K in IWN3 (indicated in brown). Illustration of the shrinkage of the HRS(HGRS-1) microdomain upon the introduction of D1657K in IWN3 (indicated in green). Each data point is an individual worm, error bars indicate Mean with 95% CI, ANOVA statistical analysis done in Prism with $p < 0.5 = *$, $p < 0.01 = **$, $p < 0.001 = ***$, $p < 0.0001 = ****$. Scale bars are 5 microns in whole coelomocyte images.

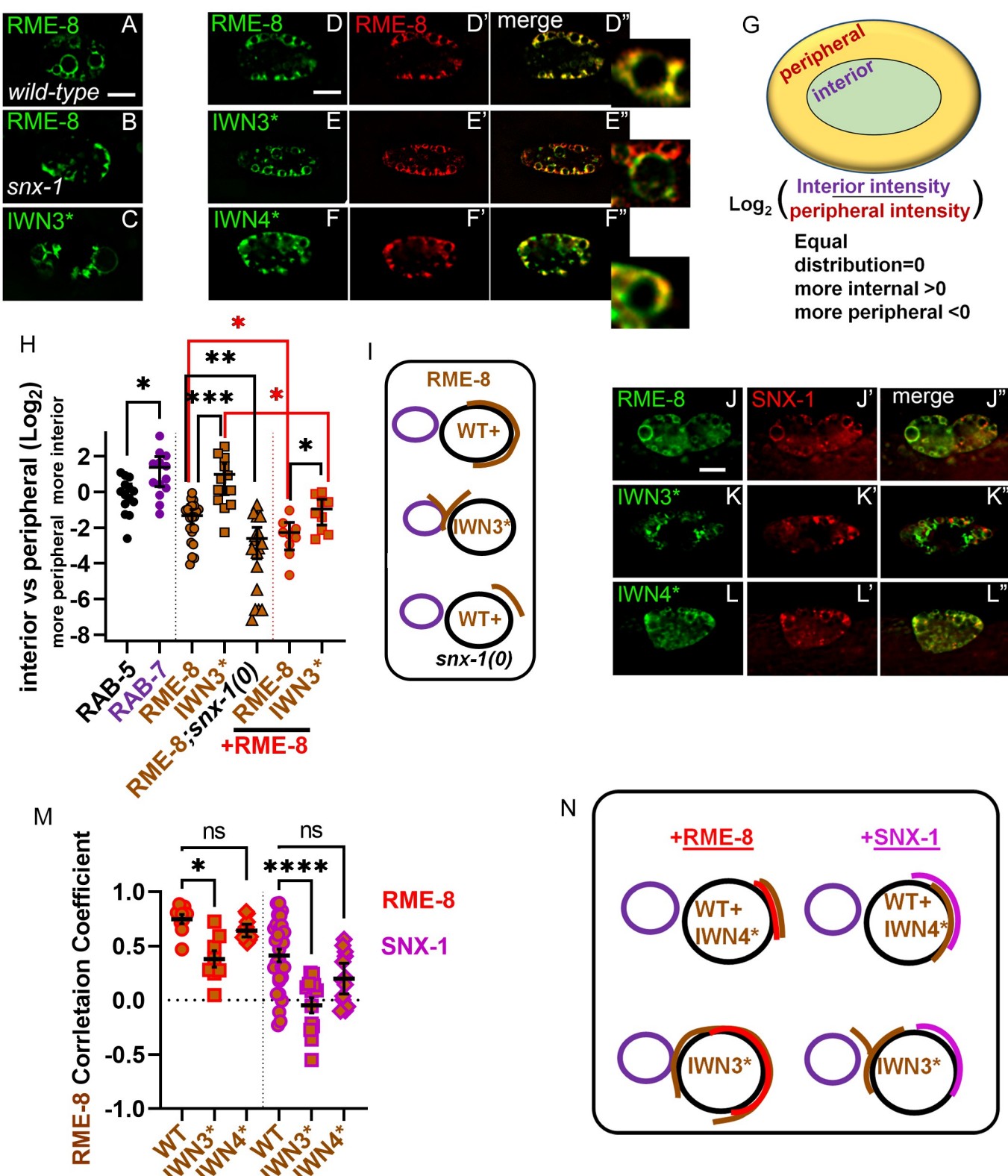

**Fig 6. SNX-1, IWN3\* and RME-8 expression levels control RME-8 microdomain localization and positioning.** (A-B) Micrograph of pCUP-4::GFP::RME-8+ wild-type (A) or (B)*snx-1(0)* mutant animals. (C) Micrograph of pCUP-4::GFP::RME-8 D1657K (IWN3\*) in wild-type animals. (D-F") Micrograph of TagRFP::RME-8 co-expressed

with RME-8+, IWN3* or IWN4* in coelomocytes. Single channels and the merge are displayed, with an inset of an endosome from the merged channels. (G) Illustration of how to calculate the interior versus peripheral labeling of RME-8 (H) Quantification of the ratio of intensity of interior versus peripheral endosome labeling of RAB-5, RAB-7 (from Fig 2B and 2C) and the indicated GFP::RME-8 allele from A-F″ plotted on a Log2 scale. A value of 0.0 indicates intensity is equal between the peripheral and interior regions. A two-fold increase in interior intensity would give a value of 1.0 and a two-fold decrease would give a value of -1.0. See illustration in G. (I) Illustration of the interior shift of the RME-8 microdomain that occurs upon the introduction of D1657K in IWN3 (indicated in brown). (J-L″) Micrograph of TagRFP::SNX-1 co-expressed with RME-8+, IWN3* or IWN4* in coelomocytes. Single channels and the merge are displayed (M) Quantification of colocalization of tagRFP::RME-8 (From D-F″) or tagRFP::SNX-1 (from J-L″) with GFP::RME-8 WT, IWN3* and IWN4*. (N) Illustration of the peripheral or interior shift of a wild-type RME-8 microdomain (indicated in red) that occurs upon coexpression with GFP::RME-8(+) or GFP::RME-8 D1657K (IWN3*), (all indicated in brown). Illustration of the tagRFP::SNX-1 (indicated in magenta) colocalization that occurs upon coexpression with the introduction of GFP::RME-8(+), GFP::RME-8 D1657K (IWN3*) (all indicated in brown). Each data point is an individual worm, error bars indicate Mean with 95% CI, student t-test statistical analysis done on ratiometric data, and ANOVA on Pearson's correlation data, with $p < .5 = *$, $p < .01 = **$, $p < .001 = ***$. Scale bars are 5 microns in whole coelomocyte images.

self-interactions occurring via homodimers rather than self-interaction within one molecule.

## RME-8 regulates SNX-1 dynamics

RME-8 and SNX-1 are binding partners, and both potentiate retrograde recycling and the separation of degradative and recycling microdomains on sorting endosomes [2,4]. The optimal SNX-1 binding domain of RME-8 encompasses IWN3 and IWN4 regions (Fig 4A). Indeed, without IWN3r, IWN4r, or both regions, RME-8 and SNX-1 display reduced overlap and appear disengaged (Fig 3D and 3F, quantified in Fig 3G, and S3A–S3E″ Fig, quantified in S3F Fig, illustrated in Fig 3N) as RME-8 lacking these sequences does not segregate into a microdomain, but rather spreads throughout the endosome (Fig 2D–2H, quantified in Fig 2L, illustrated in Fig 2M). In mammalian cells when RME-8 is depleted by siRNA, mammalian Snx1 accumulates on endosome associated membrane tubules [31]. Similarly, we observe SNX-1 overaccumulation on endosomes in *rme-8(b1023ts)* mutants at restrictive temperature (Fig 7A and 7B, quantified in Fig 7H). Unlike IWN3* and WT, none of the RME-8 domain deletion mutants can rescue SNX-1 overaccumulation in *rme-8ts* mutant animals (Fig 7C–7G, quantified in Fig 7H). Moreover, the size of SNX-1 microdomains are diminished rather than enlarged in the RME-8(IWN3*) expressing animals (Fig 7B, 7C, and 7F, quantified in Fig 7I). Not only does SNX-1 over accumulate in the absence of RME-8, but recovery after photobleaching of GFP-SNX-1 on endosomes is much slower than wild-type, never recovering to the levels found in animals expressing wild-type RME-8 within the time-frame monitored (Fig 7J–7K″, quantified in Fig 7L). These results suggest that SNX-1 may be a substrate for RME-8/Hsc70 assembly/disassembly activities, in addition to acting as an activator of RME-8 uncoating activity toward the degradative microdomain. This interpretation is supported by previous studies from our lab showing that active site mutations in the RME-8 DNAJ produce dominant SNX-1 aggregation phenotypes similar to the *rme-8 ts* loss-of-function mutant [2]. Alternate explanations remain possible however, such as loss of RME-8 leading to increased available membrane binding sites for SNX-1, leading to more stable SNX-1 binding to the endosome.

## Discussion

While endosomal microdomains are well-known features associated with the ability of endosomes to sort incoming cargo, remarkably little is understood about how such microdomains are formed, maintained, and balanced (for review see [1]). We previously identified RME-8 as a key protein in this process, acting to keep recycling and degradative microdomain separated and in balance [1,2–4]. In particular RME-8 and SNX-1 are required to limit the assembly of the opposing degradative microdomain. The pair likely act by disassembling ESCRT-0 complexes that encroach into the recycling domain where they could interfere with recycling. Here

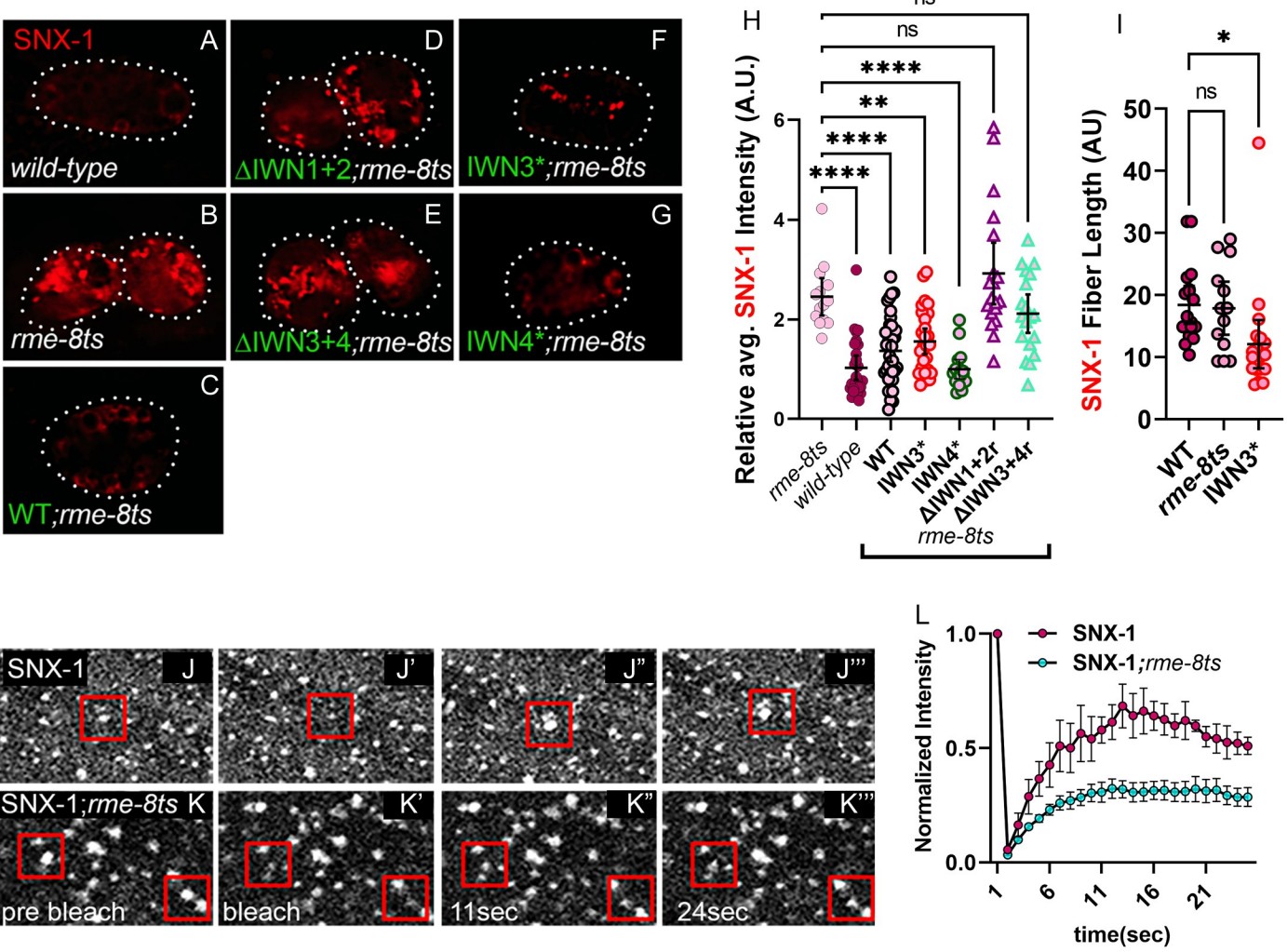

**Fig 7. SNX-1 is a target of RME-8 activity.** (A-G) Micrograph of tagRFP::SNX-1 co-expressed with GFP::RME-8 WT, N-terminal and C-terminal truncations, as well as the IWN3* and IWN4*point mutants in *rme-8ts* animals. (H) Quantification of tagRFP::SNX-1 average intensity displayed in A-G. (I) Quantification of the fiber length tagRFP::SNX-1 microdomains in animals expressing GFP::RME-8, no RME-8 and GFP::IWN3* in *rme-8ts* mutant animals. (J-K'") FRAP analysis of GFP-SNX-1 positive endosomes expressed in the hypodermis of *wild-type* and *rme-8ts* animals. Red squares indicate endosomes that were bleached. (J) Quantification of FRAP analysis of GFP-SNX-1 positive endosomes expressed in the hypodermis of *wild-type* and *rme-8ts* animals. p<0.0001. ANOVA statistical analysis done in Prism, p<0.5 = *, p<0.01 = **, p<0.001 = ***. Scale bars are 5 microns in whole coelomocyte and hypodermis subcellular images.

we sought to better understand how the large and complex RME-8 protein functions using structure-function analysis. In this study we define a role for sequences N-terminal to the DNAJ domain in endosomal recruitment. We also extend our previous model suggesting that an interaction between the DNAJ domain and sequences C-terminal to it regulate RME-8 activity.

## Predicted structure of RME-8 via AlphaFold

During the late phases of completing this manuscript, a new highly lauded AI-based system for protein structure prediction, called AlphaFold, was released. Thus, we sought to analyze the AlphaFold predicted structure for *C. elegans* RME-8 (see https://AlphaFold.ebi.ac.uk/entry/G5ED36) with respect to our structure-function results [32]. The predicted structure

shows an N-terminal domain that is enriched in beta-sheets (Fig 8H and see below). Outside of the N-terminal first 400 amino acids, much of the predicted RME-8 structure consists of a series of 5 alpha-solenoids containing HEAT repeats, with short linkers between them (Fig 8A–8D). Remarkably, these linkers correspond to the RME-8 IWN domains (IWN domains denoted in blue Fig 8A and 8B and 8E-8G). AlphaFold also predicts that the small DNAJ domain protrudes from RME-8, extending from the 3rd alpha-solenoid back toward the 2nd alpha-solenoid, positioned adjacent to IWN2 (Fig 8A and 8B). We colored the predicted 3-dimensional structure according to our pan-Eukarya conservation analysis from Figs 1 and S1 (Fig 8C and 8D). This conservation analysis indicates that in addition to IWN2, the regions outside of, but adjacent to, IWN2 and DNAJ might contribute to this DNAJ domain orientation (see light brown regions in Fig 1B and arrows Figs 1C and S1). Moreover, a region encompassing solenoids 3, 4 and part of 5 is highly conserved (Fig 9C).

## RME-8 N-terminus has three PH-like domains rather than one

Previous work indicated that human RME-8 contains an N-terminal PH-like domain, similar to the PH-domain of FERM1 [10] (Fig 8I). This domain encompassing the first 100 amino acids of RME-8 was shown to preferentially bind membranes enriched in PI(3)P and PI(3,5) P2, phosphoinositides well known for recruiting peripheral membrane proteins to early and late endosomes [10]. Our new *in vivo* structure-function analysis indicated the additional adjacent 300 amino acids in RME-8 beyond this first region that contribute to endosome recruitment (Fig 2G).

While not apparent in the primary sequence, using the Pymol [33] align feature to analyze of the AlphaFold predicted structure, we identified two additional PH-like domains within the first 400 amino acids of RME-8 (Fig 8H–8I). All three predicted PH-like domains, including the previously described domain at the extreme N-terminus, are structurally similar to the FERM1 PH domain (Fig 8H–8I). These addition sequences are absent in our RME-8 ΔIWN1 +2 transgene, hence a requirement for the two additional predicted PH-like domains provides a simple explanation as to why RME-8 ΔIWN1+2 which retains the first 100 amino acids is deficient in membrane recruitment (Fig 2G quantified in Fig 2L). Taken together these results suggest three PH-like domains may work in concert to direct RME-8 to the endosomal membrane via phosphoinositide lipid binding.

## IWN motifs as conformational control points in autoinhibition

As described above, the AlphaFold prediction suggests that the bulk of RME-8 consists of a series of 5 alpha-solenoids with short linkers between them (Fig 8A). These linkers between solenoids are predicted to lie within the IWN motifs, where they are predicted to occur as short beta-strands that terminate an alpha-helix (Fig 8A, 8D and 8E). The IWN tryptophan residue is predicted to be buried and surrounded by hydrophobic residues (Fig 8F). The isoleucine and asparagine, however, are predicted to be on the opposite side of the beta strand, both interacting with nearby peptide backbone features. We find that all four IWNs display similar predicted secondary and tertiary structure (Fig 8D). In the case of IWN3, the aspartic acid 1657 that is mutated to lysine in IWN3*, is predicted to interact with the adjacent alpha-helix (Fig 8G). A substitution of lysine at this position would likely be quite disruptive to the orientation of the third and fourth solenoids.

The prediction that the conserved IWN repeats represent linkers between domains supports the idea that the IWN motifs control large scale conformational changes in RME-8. This is consistent with our new data that sequences in RME-8 C-terminal to the DNAJ domain contain autoinhibitory activity, especially our finding that a single point mutation in IWN3 is

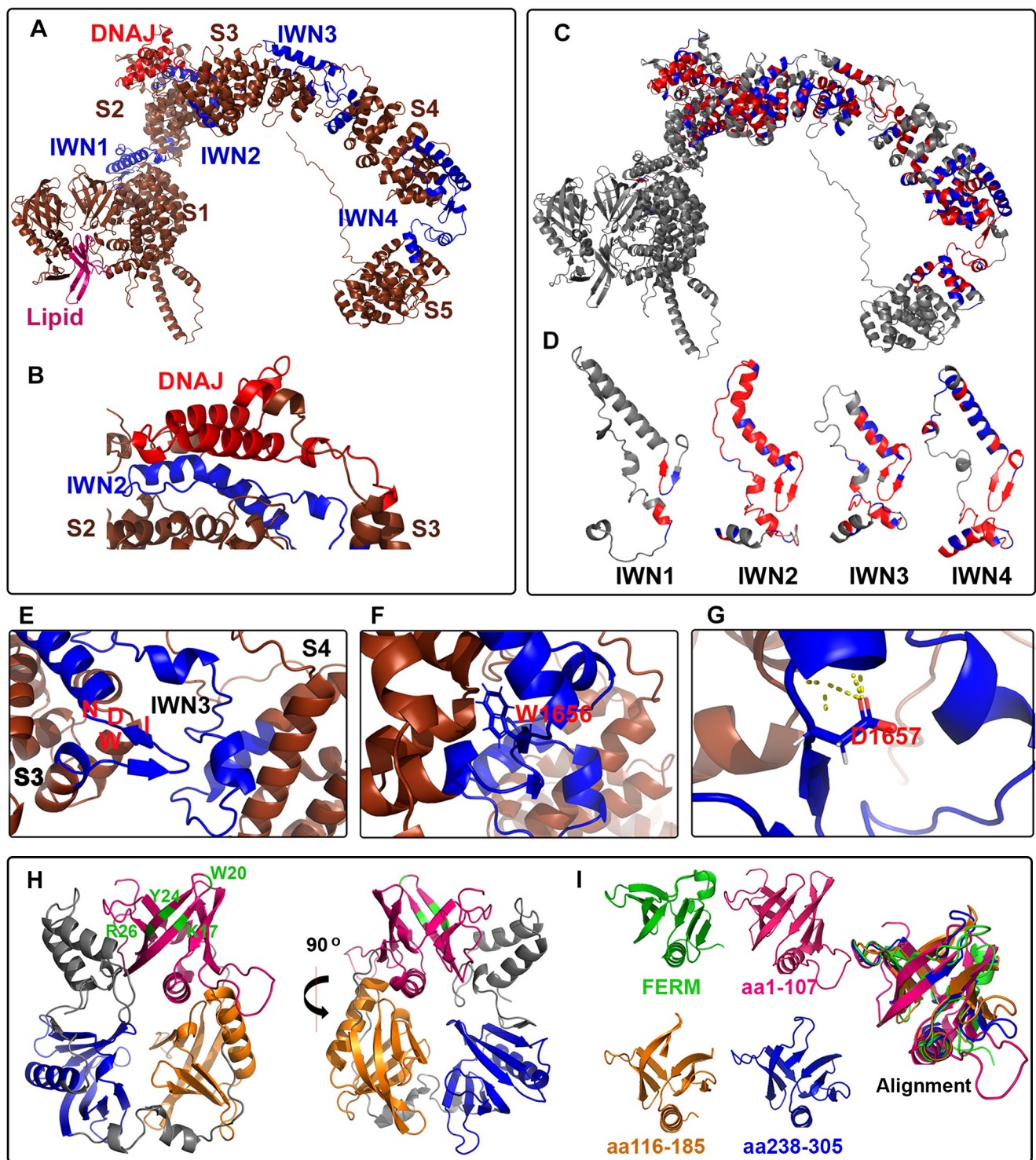

**Fig 8. RME-8 AlphaFold structural predictions: conformational control mediated by IWN motifs and lipid binding mediated by three PH-like domains.**
(A) Ribbon diagram of AlphaFold generated PDB file. The previously defined lipid binding domain is depicted in magenta. IWN domains defined by Zhang et al 2001 are depicted in blue. The DNAJ domain is depicted in red. The five alpha-solenoid regions are labeled S1-S5. (B) A zoomed in view of the juxtaposition of IWN2 and the DNAJ domain, with portions of alpha-solenoid 2 and alpha-solenoid 3 that are separated by the IWN2 elbow-like structure. (C) ribbon diagram demonstrating the conserved residues from Figs 1 and S1. Highly conserved residues are colored in red, moderately conserved residues are colored in blue. (D) Using the same color scheme as panel C, the IWN domains are represented to show their similarity in form as well as conservation across Eukarya. (E) A zoomed in view of the IWN3 linker between alpha-solenoid 3 and alpha solenoid 4. The position of the (I) Isoleucine, (W)Tryptophan, (D) Aspartic acid, and (N) Asparagine of the IWN3 domain are indicated in red. (F) The position of W1656 of the IWN3 domain is predicted to be buried, and on

the opposite face of the Beta-strand as I, D and N. (G) The position of D1656 which is mutated to lysine in IWN3* in the predicted structure. (H) Ribbon diagram of the first 400 amino acids of the AlphaFold generated structure. The previously defined domain is indicated in magenta, with key residues for phospholipid binding indicated in green. The two additional PH-like domains are indicated in blue and orange. (I) Ribbon diagrams of the FERM domains used to align to the three beta-strand rich regions of the N-terminus in green. The three PH-like domains of the RME-8 N-terminus are colored as in (H). An alignment of all four Beta-strand rich regions is shown to the right of the individual domains.

sufficient to produce a hyperactive protein, and that IWN3 and IWN4 regions of the protein physically interact with the DNAJ domain. We currently favor a model in which the relevant autoinhibitory interactions between IWN3-IWN4 and the DNAJ domain occurs between RME-8 molecules in an oligomer, rather than within a single RME-8 molecule. In particular, our finding that wild-type RME-8 can inhibit the hyperactivity and alter the microdomain positioning of the IWN3* mutant version of RME-8 could be explained if oligomerized RME-8 molecules are autoinhibited.

## The role of SNX-1

We have previously proposed that SNX-1 is a positive regulator of RME-8 as loss of either SNX-1 or RME-8 has similar effects on cargo recycling and expansion of the degradative microdomain [2]. Our model predicts that SNX-1 binding derepresses RME-8, allowing RME-8 to uncoat both ESCRT-0 and, intriguingly, SNX-1 itself. If oligomerization of RME-8 inhibits activity, then SNX-1 binding could act to release active RME-8 monomers that can work with Hsc70 to disassemble both SNX-1 and ESCRT-0 complexes (Fig 9B).

According to this model, when the RME-8/DNAJ interaction is reduced, as in the case of IWN3*, more RME-8 can de-oligomerize and shift its positioning to invade the degradative microdomain, leading to the hyperactive uncoating of HRS/HGRS-1 that we observe with IWN3*(Fig 9B and 9D). As with IWN3*, IWN4* also reduces RME-8/DNAJ interaction, but the interaction with SNX-1 is different. IWN4* improves the interaction with SNX-1 and is *not* hyperactive suggesting that release of SNX-1 maybe an important part of the activation cycle of RME-8.

Results are also different when large portions of the C-terminal regions encompassing IWN3 are fully deleted, a situation where we expect that RME-8 also cannot bind to SNX-1 or self-oligomerize. In this case we do observe spreading of nonfunctional RME-8 around the endosome (Fig 9C). This difference between the IWN3* point mutation and full domain deletion indicates another essential function of RME-8 sequences after the DNAJ domain beyond autoinhibition.

In addition to stimulating Hsc70 ATPase activity, substrate binding is a typical function of DNAJ domain proteins. Because DNAJ-domain cochaperones typically act as cargo adapters for Hsc70, the RME-8 C-terminus is likely binding Hsc70 substrate(s). It is still unclear what the direct substrate of RME-8/Hsc70 is that controls the degradative microdomain. Candidate substrates include ESCRT-0 components HGRS-1/Hrs or STAM, Clathrin that associates with ESCRT-0, or other associated molecules, but clear identification of the key substrate awaits further studies.

New data presented in this work, along with published observations on RME-8 siRNA phenotypes in mammalian cells, also suggests that SNX-1 itself is a good candidate to be a substrate for RME-8/Hsc70 chaperone activity. Observations as to the importance of RME-8 for control of endosomal SNX-1 has been previously reported by Freeman et al. [31]. This work showed that relatively static Snx1 coated endosomal tubules accumulated after treatment of Hela cells with RME-8 siRNA. Consistent with these findings, we show that SNX-1 accumulates on endosomes in an *rme-8ts* mutant, a phenotype that is rescued by IWN3* but not by

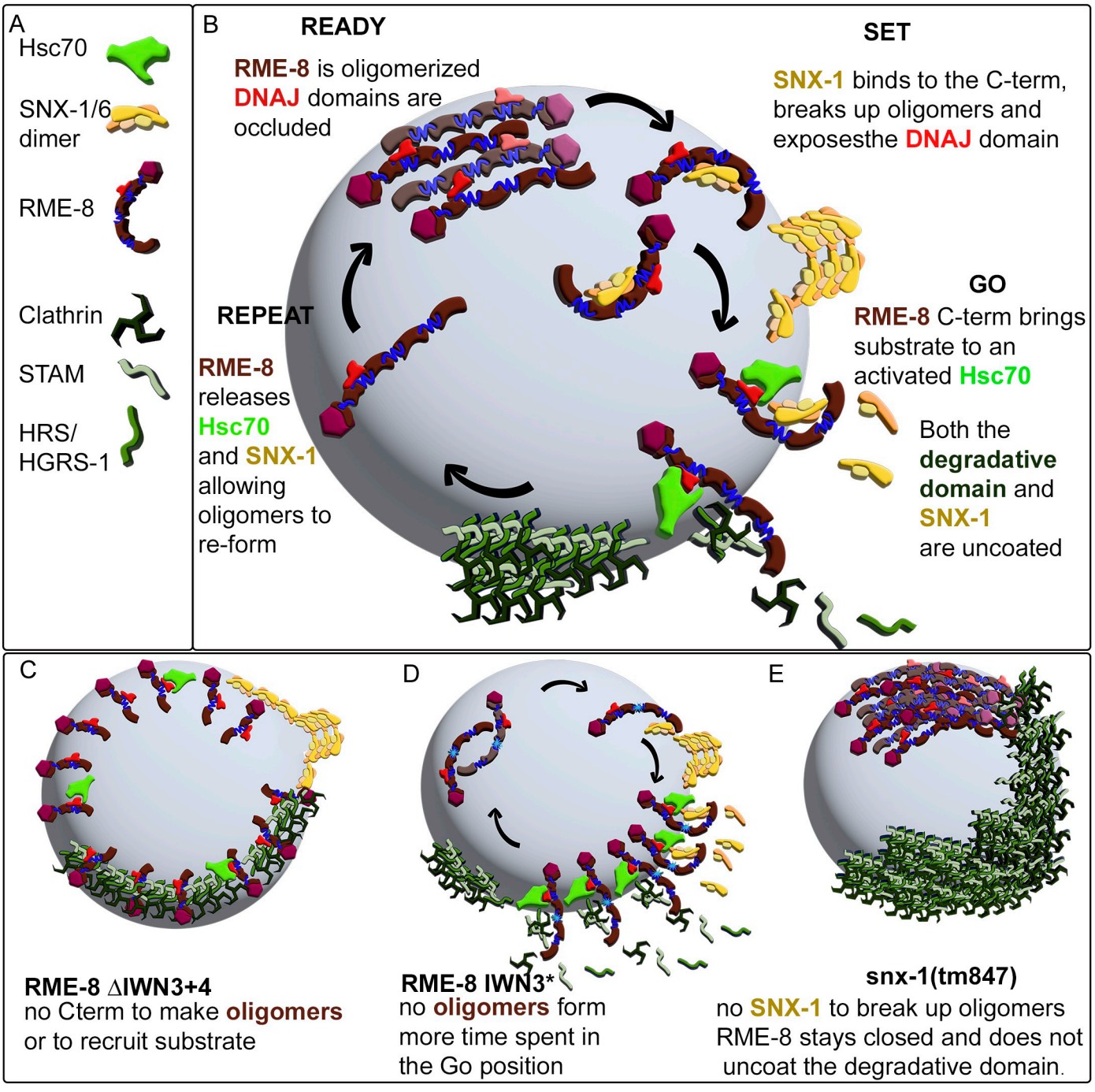

**Fig 9. Ready Set Go model for cyclical activation of RME-8.** (A) Legend (B) Illustration of the Ready Set Go Repeat model of cyclical RME-8 activation on the endosome. We propose that RME-8 exists in a few different states in the cell; an oligomerized inactive form with the DNAJ domain occluded (Ready), an open form bound to SNX-1 with the DNAJ domain exposed (Set). An active form bound by Hsc70 and substrate (Go). This model is informed by the idea that the RME-8 activator SNX-1 is also a target of RME-8 uncoating activity, allowing the process to be cyclical. (C) An illustration of the spread of RME-8 and accumulation of SNX-1 and HRS/HGRS-1 in coelomcytes expressing RME-8 C-terminal truncations. (D) An illustration of the model for how the IWN3* mutant disrupts RME-8 oligomers independently of SNX-1 allowing for more RME-8 to be in the active "go" state. (E) Illustration of the proposed HRS/HGRS-1 and RME-8 oligomeric accumulation that occurs in *snx-1(0)* animals as observed in previous studies [2].

defective RME-8 truncation transgenes (Fig 7). Additionally, FRAP analysis of SNX-1 positive endosomes shows that recovery of GFP::SNX-1 is dramatically slowed in the absence of RME-8 (Fig 7). RME-8 activation by SNX-1 could be transient if the activation of the RME-8 DNAJ-domain ultimately terminates interaction of SNX-1 with RME-8, causing reversion of RME-8 to the autoinhibited state (Fig 9). This strategy of activation of a powerful regulator tied to inactivation is a common theme in membrane trafficking. Indeed SNX-1 being required for RME-8 activation as well as a target of RME-8 disassembly activity could effectively localize the powerful Hsc70 activity to the proper place and time.

We envision the SNX-1/RME-8 network in the recycling microdomain as part of a larger system that acts to create and separate functional microdomains on endosomes, with likely feedback regulation going in both directions. More work will be required to understand how the protein complexes within microdomains act to balance the activities of the endosome to allow correct sorting of incoming molecules and rebalance such activities as loads and cargo types change over time. The *C. elegans* coelomocyte provides an excellent system to answer such questions.

## Materials and methods

All *C. elegans* strains were derived originally from the wild-type Bristol strain N2. Worm cultures, genetic crosses, and other *C. elegans* husbandry were performed according to standard methods [34]. A complete list of strains used in this study can be found in S1 Table.

### Yeast two-hybrid mutagenesis screen

We used error prone PCR [35] of RME-8 cDNA encoding amino acids 1322–2279 (J-domain to C-terminus) in PDEST32, followed by gap repair transformation. Mutagenized RME-8 fragments were co-transformed with a gapped vector, that included 100 bp homology arms, into JDY27 yeast expressing SNX-1 amino acids 221–472 (BAR domain) in PDEST22. JDY27 is a yeast strains that includes several genomic reporter genes dependent on the Y2H interaction, including ADE-2 and LacZ color assays, as well as HIS-3 and URA-3 growth assays (ade2-101 trp1-901 leu2-3.112 his3Δ200 ura3-52 gal4Δ gal80Δ SPAL::URA3 LYS2::GAL1-HIS GAL2-ADE2 met2:GAL7-LacZ (or GAL1-LacZ) can1R alpha). Approximately 10,000 transformed yeast were then plated on SC-HIS-LEU-TRP media and allowed to grow for 3 days at 30˚C. These colonies were then replica plated to selective media SC-HIS-LEU-TRP-URA and allowed to grow for 10 days at 30˚C. The 104 colonies that displayed an initial URA+ phenotype were patched onto SC-HIS-LEU-TRP-URA and selected for robust growth after 3 days at 30˚C. We then retransformed the plasmids recovered from these colonies into the JDY27 SNX-1 strain to ensure the URA+ phenotype was dependent on the RME-8 plasmid. 30 colonies were then sequenced for mutations in the 1322–2279 region. Most colonies had multiple mutations, therefore we chose the strongest URA+ colonies to pursue, introducing mutations singly into PDEST32 RME-8 plasmids by site directed mutagenesis. We identified mutations E1962K and N1966K from two different colonies and chose to pursue their function further.

### Yeast transformation and plasmid recovery

Yeast transformations were performed using cells grown in liquid 1x YPD overnight at 30˚C. The pellet was then washed 1x with distilled water and resuspended in a solution of 0.1M LiOAc + 1x TE, at a 1:2 pellet size to buffer volume. 100ul of yeast slurry was then combined with 1μg of DNA and 10ul of denatured 10mg/ml salmon sperm DNA. Yeast were then incubated at 30˚C for 30 minutes before adding 1ml 44% PEG + 0.1M LiOac and 1x TE, after which it was allowed to incubate for another 30 minutes at 30˚C. Following this, DMSO was

added at a 1:10 ratio to the PEG, and yeast were heat shocked in a 42˚C water bath for 13 minutes. After the heat shock, yeast were spun down in a microcentrifuge and resuspended in 50µL 5mM $CaCl_2$ before being plated onto selective media.

Yeast plasmid recovery performed using QIAprep Spin Miniprep Kit and a user adapted Protocol by Michael Jones, Chugai Institute for Molecular Medicine, Ibaraki, Japan. Essentially a colony of yeast is resuspended in 250ul of P1 buffer combined with 100 µl of acid-washed glass beads (Sigma G-8772) and vortexed for 5 min. The slurry is then used in the typical miniprep protocol, with the final eluate of the column used to transform bacteria for amplification of the plasmid DNA.

## Homology searches

We used the NCBI BLAST to probe the over 4000 proteomes in the NCBI database for the existence of DNA J proteins that also contained IWN repeats. We used the *C. elegans* RME-8 amino acid sequence and the default BLAST algorithm to probe each taxonomic group for an RME-8 homolog. We filtered the results with a coverage threshold of >40% and e value < .001. We then hand curated each positive to ensure that it contained IWN repeats defined by I/L/V-W- ζ ζ (hydrophylic amino acids), all of the hits with at least 40% coverage contained at least three of the IWN repeats.

## Display of tree of life

To display the phylogenetic conservation of RME-8 we used a list of TAXID's from the 4001 sequenced genomes from [36] the to create a phylogentic tree using phyloT. phyloT then generated a phylogenetic tree based on the NCBI taxonomy. The taxonomic tree branch order was then uploaded to iTOL version 5.7 [37] which is based on Tree of Life v1.0 [38]. We then used iTOL to illustrate the branches that contained or did not contain an RME-8 homolog [37,38].

## Single copy MiniMos transgenes

Plasmids for *C. elegans* coleomocyte expression were produced by standard methods including *in vitro* recombination via the Gateway system (ThermoFisher) and/or Gibson Assembly using the Nebuilder system (NEB). Plasmid backbones were pCFJ1662 and pCFJ910 (Addgene) and used promoters from the *snx-1* or *cup-4* genes. Our minimos protocol for single copy transgene integration is based on a protocol found on wormbuilder.org as described in [39]. The gonad arms of first day gravid adults were microinjected with plasmid mixtures including 10ng/ul drug resistant expression plasmid (G418 or Hygromycin) [40,41] 10ng/µl pGH8, 2.5ng/µl pCFJ90, 65ng/µl pCFJ601, and 10ng/ul pMA122. 2–3 injected animals per plate were incubated at 25˚C, with selection drug added between 24–72 hrs post injection. Plates were screened for single copy integrated transformants after a minimum of 10 days of growth, focused on populations that survive drug selection and lack extrachromosomal arrays visualized with the mCherry co-injection markers. Candidate single-copy integrants were passaged on drug containing plates and analyzed for expression. Lines displaying 100% transmission of the expressed transgene without drug selection were frozen and used for experiments.

## Microscopy and image analysis

Live animals were mounted on 10% agarose pads and immobilized using 0.1um polystyrene microspheres (Polysciences 00876–15) [42]. Multiwavelength fluorescence Z-series with 0.2 micron step size were obtained using a SOLA SE 365 Solid state light engine, Axiovert 200M (Carl Zeiss MicroImaging) microscope equipped with a digital CCD camera (QImaging;

Rolera EM-C2), or Axio Observer (Carl Zeiss MicroImaging) microscope equipped with a scientific sCMOS camera (Photometrics Prime 95B), captured using MetaMorph 7.7 software (Universal Imaging) and then deconvolved using AutoDeblur X3 software (AutoQuant Imaging). Colocalization analysis was done using MetaMorph 7.7 colocalization plugin, whereby intensities in each channel were thresholded and Pearson's correlation was analyzed. Intensity measurements were also analyzed by MetaMorph 7.7, using thresholded images captured under identical acquisition parameters. FRAP was done as described in [4].

### Measuring membrane to cytoplasm ratio

One coelomocyte per animal and at least 5 animals per genetic background were measured. Each data point on the graph is from one animal. In each coelomocyte 6 different 3 micron diameter circular regions were measured, three of which encompassed an endosome and three of which encompassed non-endosomal portions of the coelomocyte. The average of the intensity of the three regions which encompassed an endosome was divided by the average of the three regions that did not encompass an endosome to arrive at the ratio.

### Measuring Microdomain spread

Line scans around the circumference of at least three endosomes per coelomocyte were drawn. The fraction of the line where the intensity was above a threshold was measured, this number indicated the relative size of the microdomain on the endosome. Each data point is the average of at least three endosomes per coelomocyte in one animal.

### Measuring peripheral versus internal intensity

The coelomocyte was divided into two equally sized regions, on that encompasses the peripheral section and one that encompasses the internal portion. The intensity above threshold in each region was measured. The internal intensity was divided by the peripheral intensity to arrive at a ratio, which was plotted on a log2 scale. A value of 0.0 indicates intensity is equal between the peripheral and interior regions, a value of 1 indicates the interior intensity is two fold greater than peripheral, conversely a -1 value indicates a two-fold greater peripheral intensity, would see illustration in Fig 6G.

### Graphing and statistics

Results were all graphed using GraphPad Prism 8.4.3 software and significance was measured by one-way ANOVA or students t-test.

### Structure Prediction

A FASTA file was generated containing the C. elegans RME-8 amino acid sequence and used as input for structure prediction by Alphafold 2.1.1 [32]. The resulting PDB files were visualized in Pymol [33], which was additionally used for structural alignments of RME-8's beta barrel structures and FERM1's PH-domain of 1MIX Pymol [33]. A similar structure prediction of C. elegans RME-8 is now available on the AlphaFold Protein Structure Database [32,43].

## Supporting information

**S1 Fig. Supplement to Fig 1.** (A) Expansion of Fig 1C with more organisms included. (B) Annotated display of *C. elegans* RME-8 sequence from the multiple sequence alignment in Fig 1B. The defined lipid binding domain from [6] is indicated in pink. IWN repeats defined by [2] purple. The DNAJ domain is highlighted in orange. Light brown arrows indicate regions of

high conservation that lie outside of established functional domains. Individual amino acids are colored according to conservation across Eukarya with grey indicating no conservation, blue indicating moderate conservation and red indicating high conservation.
(PDF)

**S2 Fig. Supplement to Fig 2.** - (A-E) Micrograph of pCUP-4::GFP::RME-8 full length, N-terminal and C-terminal truncations illustrated in Fig 2J expressed in the *rme8- ts* mutant background at the restrictive temperature. Scale bars are 5 microns in whole coelomocyte images. (F) Quantification of coelomocyte size in *rme-8ts* mutant animals expressing ectopic RME-8+, N-terminal, or C-terminal truncations. (P).
(PDF)

**S3 Fig. Supplement to Fig 3.** (A-E") Micrograph of pCUP-4::GFP::RME-8 *wild-type*, N-terminal, and C-terminal truncations co-expressed with tagRFP::SNX-1 in coelomocytes of *wild-type* animals. (F) Quantification of colocalization of pCUP-4::GFP::RME-8 *wild-type* and truncations with tagRFP::SNX-1 in coelomocytes of *wild-type* animals. (G-K") Micrograph of pCUP-4::GFP::RME-8 *wild-type*, N-terminal, and C-terminal truncations co-expressed with tagRFP::HRS(HGRS-1) in coelomocytes of *wild-type* animals. (L) Quantification of colocalization of GFP::RME-8 *wild-type* and truncations with tagRFP::HRS(HGRS-1)in *wild-type* animals. (M) Quantification of HGRS-1 intensity in animals expressing pCUP-4::GFP::RME-8 *wild-type* and truncations.
(PDF)

**S4 Fig. Supplement to Fig 4.** (A) Empty vector or SNX-1 (aa221-472) expressed in pDEST22 tested for interaction with of the RME-8 C-terminus with DNAJ domain. The RME-8 fragments were expressed in pDEST32 using JDY27 containing URA3, ADE2, and HIS3 reporter genes. 5ul of suspended yeast at 1, and 0.1 OD's were spotted on SC-LEU-TRP growth, or assay plates SC-LEU-TRP -URA. Lysines and Arginines of helix II of the DNAJ domain were targeted for doped oligo mutagenesis and selected for increased interaction with SNX-1. The aa1322-2279 fragment with lysines at position 1347 and 1356 substituted with glutamic acid was sufficient to show growth on assay media when combined with SNX-1 BAR domain, singly these mutations were not able to grow on the assay media. Arginines 1342 and 1342 were also substituted with glutamic acid and failed to grow well on the assay media. (B) Empty vector or SNX-1 (aa221-472) expressed in pDEST22 tested for interaction with of the RME-8 C-terminus with our without its DNAJ domain, and mutants isolated in the screen. The RME-8 fragments were expressed in pDEST32 using JDY27 containing URA3, ADE2, and HIS3 reporter genes. 5ul of suspended yeast at 1, and 0.1 OD's were spotted on SC-LEU-TRP growth, or assay plates with increasing stringency; SC-LEU-TRP-HIS, SC-LEU-TRP-HIS +25mM 3AT, and SC-LEU-TRP-URA. A schematic representation of the RME-8 fragments tested are illustrated to the left of the yeast two hybrid assay. (B) A schematic of the Fragment mutated represented to the left of the assay. 5ul of suspended yeast at 1, and 0.1 OD's were spotted on SC-LEU-TRP growth or SC-LEU-TRP-URA assay media.
(PDF)

**S5 Fig. Supplement to Fig 5.** (A-D) Micrograph of pSNX-1::Citrine::HRS/HGRS-1 expressed in coelomocytes of *wild-type (A) rme-8ts(b1023)* animals (B), CRISPR generated *rme-8 E1962K* animals (C), and heterozygous CRISPR generated *rme-8 (E1962K/N1966K/E1967K)* animals(D). (E) Quantification HRS/HGRS-1 average intensity animals represented in A-D. (F-G)) Micrograph of tagRFP::HRS(HGRS-1) in *wild-type* animals expressing pCUP-4::GFP:: RME-8+ or D1657K mutation in IWN3 (IWN3*). Quantification HRS/HGRS-1 average intensity animals represented in F-G. Each data point is an individual worm, error bars indicate

Mean with 95% CI, student t-test statistical analysis done in Prism with $p < .5 = $ *, $p < .01 = $ **, $p < .001 = $ ***. Scale bars are 5 microns in whole coelomocyte images.
(PDF)

**S1 Tablestylefix. Strains used in this study.**
(PDF)

**S1 Data. Data for Fig 2K.**
(TXT)

**S2 Data. Data for Fig 2L.**
(TXT)

**S3 Data. Data for Fig 3G.**
(TXT)

**S4 Data. Data for Fig 3H.**
(TXT)

**S5 Data. Data for Fig 3M.**
(TXT)

**S6 Data. Data for Fig 5D.**
(TXT)

**S7 Data. Data for Fig 5E.**
(TXT)

**S8 Data. Data for Fig 6H.**
(TXT)

**S9 Data. Data for Fig 6M.**
(TXT)

**S10 Data. Data for Fig 7H.**
(TXT)

**S11 Data. Data for Fig 7I.**
(TXT)

**S12 Data. Data for Fig 7L.**
(TXT)

**S13 Data. Data for S2F Fig.**
(TXT)

**S14 Data. Data for S3F Fig.**
(TXT)

**S15 Data. Data for S3M Fig.**
(TXT)

**S16 Data. Data for S3L Fig.**
(TXT)

**S17 Data. Data for S5E Fig.**
(TXT)

## Acknowledgments

We would like to thank the Grant lab for critical comments and suggestions.

## Author Contributions

**Conceptualization:** Anne Norris, Barth D. Grant.

**Data curation:** Anne Norris.

**Formal analysis:** Anne Norris, Collin T. McManus, Simon Wang, Ruochen Ying.

**Funding acquisition:** Anne Norris, Barth D. Grant.

**Investigation:** Anne Norris, Collin T. McManus, Simon Wang, Ruochen Ying.

**Methodology:** Anne Norris.

**Project administration:** Anne Norris.

**Resources:** Anne Norris.

**Supervision:** Anne Norris, Barth D. Grant.

**Validation:** Anne Norris.

**Visualization:** Anne Norris.

**Writing – original draft:** Anne Norris.

**Writing – review & editing:** Anne Norris, Collin T. McManus, Barth D. Grant.

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
