## [Decision Letter · Decision Letter 0]

21 Jul 2022

Dear Dr Norris,

Thank you very much for submitting your Research Article entitled 'Mutagenesis and structural modeling implicate RME-8 IWN domains as conformational control points' to PLOS Genetics.

The manuscript was fully evaluated at the editorial level and by independent peer reviewers. The reviewers appreciated the attention to an important topic but identified some concerns that we ask you address in a revised manuscript. We therefore ask you to modify the manuscript according to the review recommendations. Your revisions should address the specific points made by each reviewer.

[LINK]

Yours sincerely,

Ken Sato

Guest Editor

PLOS Genetics

Gregory P. Copenhaver

Editor-in-Chief

PLOS Genetics

Reviewer's Responses to Questions

**Comments to the Authors:**

Reviewer #1: Norris et al. present the first molecular hints to how the large RME-8 protein regulates endosomal microdomain formation. Using an evolutionary analysis, they identify new conserved domains in the protein and lay the groundwork for years if not decades of new discoveries. They delve deeper into the function of the IWN domains, revealing key roles of the third and fourth IWN domains with deletion and point mutants as molecular hinges regulating physical interactions with the DnaJ domain and SNX-1. Finally, they propose a mechanistic model of RME-8 regulation, informed by their structure-function data as well as AlphaFold structure predictions. This study represents an important step forward for understanding microdomains and is likely to be highly impactful given the breadth of information that it reveals.

Major suggestions

All searches for RME-8 homologs in Fig. 1 were performed using RME-8 from a single species. Would more divergent homologs be identified by searching with more closely related species? For example, if you search fungal proteomes with a sponge or choanoflagellate RME-8 homolog, do you still find nothing passing the criteria? Similarly, searching gymnosperm proteomes with an angiosperm RME-8 or alveolate proteomes with an amoeba RME-8 would strengthen the claim that RME-8 is absent in these lineages.

In Figure 2G&J, the constructs are labeled ∆N-term, but Fig. 2E, H & I are labeled ∆IWN1+2. Considering that these large deletions also remove the newly identified conserved domains in addition to the IWN domains, it would be helpful to include the new yellow domains in the diagrams in Fig. 2G (or better using a to-scale block cartoon, the cartoons from Fig. 9, or the AlphaFold models) and to be consistent in the naming throughout the figure. Similarly, the discussion of the C-terminal deletion should be more cautious about assigning the microdomain function to IWN3/4 in lines 156-159, given the other domains in the deletion.

Line 149-151 - Figure 3D does not quantify SNX-1 spread, because it is quantifying colocalization with RME-8. That will change based on the observed changes to RME-8 localization from Fig. 2. It would be more informative to perform an analysis similar to Fig. 2I for SNX-1 spread. The similar criticism applies for using Fig. 3H to measure HRS spread in the previous sentence, although that is more carefully worded, as well as a separate analysis of RME-8 spread and HRS spread in the IWN3* and IWN4* mutants in Fig. 5 and RME-8 spread and SNX-1 spread in Fig. 6.

Fig. 4A - Has the IWN4 been tested to bind SNX-1 on its own (or a construct like aa ~1670-2279 in Y2H)? It’s strange to refer to the 1388-2279 region as the minimal domain in line 215 without evidence that removing the IWN3 sequence is detrimental to SNX-1 binding. Given the results of the D1657K experiment in Fig. 4B, the SNX-1 binding region may map into the sequence C-terminal to IWN3, such as the newly identified yellow domain. The DnaJ domain and SNX-1 seem likely to bind slightly different sites in the C-terminal region, but still sterically hinder the other.

A great test of the author’s uncoating model would be to make complementary charge mutations in the DnaJ and IWM3 domains to restore the intramolecular interaction and allow SNX-1 binding while disrupting the Hsc70 interaction. Do AlphaFold multimer predictions using separate DnaJ and IWM3-4 domains allow the authors to guess whether D1657 is likely to interact with K1347 or K1356? Then a D1657K K13**D double mutant could be informative in their Y2H and in vivo assays.

What are the effects of IWN4* on SNX-1 localization (i.e. Fig. 6-7)? It’s not clear how the authors draw a conclusion on RME-8 acting independently of SNX-1 in line 271-272 by examining a mutant that increases SNX-1 binding.

What is the effect of a snx-1 mutant on IWN3* localization? Would it be similar to wild type RME-8?

The conclusion that Hsc70 uncoats SNX-1 is tenuous based on the existing data. A direct test of whether Hsc70 uncoats SNX-1 would be to examine SNX-1 accumulation in an Hsc70 mutant (or in Hsc70-binding RME-8 mutant). An alternate interpretation for the accumulation of SNX-1 in the absence of RME-8 could be that the large RME-8 protein obscures stronger SNX-1 binding sites or lipid interactions. If all binding sites are stably occupied, that would also lead to the reduced SNX-1 mobility by FRAP. The author’s show no direct evidence for turnover and need to phrase this as one interpretation among others.

In general, it would be helpful if the AlphaFold models were discussed in the context of the individual figures, rather than saved for the discussion. For example, Fig. 8A-D would fit well in Fig. 1, while Fig. 8H-I could accompany the N-term deletion data of Fig 2. Also, how does the N-term deletion construct compare to the predicted AlphaFold structural elements like the three PH domains? The discussion hints that the deletion may have contained some of the predicted PH domains, but it is not made explicit with amino acid level comparisons.

Would the predicted lipid-binding surfaces of all three PH domains be able to interact with a flat or curved lipid bilayer at the same time, or do the authors think it more likely that they would interact in the context of an RME-8 oligomer, for example with each protein using a different PH domain to interact with the endosome membrane?

Minor suggestions

Line 65-66 lacks a reference. Or is this a finding of this study?

Line 70-72 lacks a reference. What and where were this previous evidence?

In Figure 2G, there’s enough vertical space in the figure to spell out the full proposed function in two rows. Loc could be short for location or localization. Pos could be short for positive or positioning.

Line 155-157 & line 302 - This conclusion would depend on deleting the IWN domains individually, but this data does not appear in Fig. 3.

Line 162 - More accurately RME-8 C-term, since the deletion removes other domains.

Fig. 4A - The growth in the SNX-1 assay would be easier to see if the brightness was increased in this image (more similar to 4C).

Line 209-211 - Is this referring to Fig. 4C? Y2H is rarely described as a pulldown.

Line 213 - Is Figure. 6E supposed to refer to figure 9?

Fig. 4D, the dark yellow N is hard to see in the dark blue background. Consider changing its color. Also, a similar sequence assembly for IWN3 would be helpful for interpreting the D1657K results.

Line 220-221, a conclusion re: SNX-1 interaction would also be helpful here.

Is the right side of Fig. 4G ever discussed?

Lines 266-272 - The section on the in vivo IWN4* mutant results are hidden in the IWN3* section. It would be helpful to give these their own section with a separate heading.

Fig. 6A-D are mislabeled in the figure legend. Also, in A, 1 is equal distribution. 0 would be no interior intensity.

Fig. 6I - Are any changes significant? Also, the log2 scale is confusing if the authors want us to look for <1>.

Line 282 - on = of?

The data for RME-8(+) effect on RME-8(IWN3*) localization in Fig. 6E/K is not discussed in the text, although it appears in the conclusion in line 284.

Line 288-290 - sentence fragments

The first paragraph of the discussion would be a strong intro paragraph.

Paragraph 426-432 is not relevant to any of the data reported in this manuscript and would be more appropriate in an RME-8 review.

Fig. 9B - Ready - Typo in occluded.

No methods are given for structural prediction analyses.

Reviewer #2: Endocytosed cargo can be recycled by to the membrane or targeted to the lysosome for degradation. Previous work from Norris et al (2017) demonstrated that RME-8 and SNX-1 regulate endosome recycling, colocalize on endosome microdomains and restrict the localization of the ESCRT protein HGRS-1 to a degradative microdomain. The current study focuses on gaining a mechanistic understanding of RME-8 regulation and function. They identify and characterize IWN domains as important mediators of the RME-8 regulation and function. A phylogenetic analysis demonstrates that RME-8 is an ancient protein and identifies highly conserved regions that are suggestive of important roles in protein function. Particularly IWN domains, IWN1 and IWN2 in the N-terminal half of the protein between the N-terminal lipid binding domain and the centrally located DNAJ domain and IWN3 and IWN4 in the C-terminal half of the protein. Deletions of large portions of the protein containing IWN1/2 demonstrates that it is partially required for localization to endosomes. Deletion of the C-terminus containing the IWN3/4 domains results in stronger and more uniform RME-8 localization to endosomes indicating a role maintain localization to microdomains. Both deletions disrupt RME-8’s function in maintaining HGRS-1 degradative microdomains. The IWN3/4 region interacts with SNX-1 by yeast 2-hybrid, but inclusion of the DNAJ domain blocks SNX-1 binding under stringent conditions. An error prone PCR screen identified E1962K and N1966K mutations in the IWN4 domain that permit binding of SNX-1 in the presence of the DNAJ domain. The E1962K mutation also reduces the interaction of the IWN3/4 with the DNAJ domain indicating that an intramolecular interaction between IWN3/4 and DNAJ domains restricts SNX-1 binding. A similar PCR screen also identified a K1347E/K1356E mutation in the DNAJ binding that reduces the interaction with the IWN3/4 domains and permits stronger SNX-1 binding. Interestingly, a charge reversal mutation in the IWN3, D1657K, is more strongly activated regarding restricting HGRS-1 localization on endosomes, it diminishes the interaction with the DNAJ domain, however it does not permit SNX-1 binding in the presence of the DNAJ domain. The IWN3 D1657K mutation also results in a shift in RME-8 localization from being more peripheral to more interior in the coelomocytes. Colocalization with wild-type RME-8 demonstrates the change in localization, though wild-type RME-8 does partially rescue the localization defect consistent with potential dimerization. The IWN3 mutation also has reduced localization with SNX-1 but can rescue SNX-1 localization defects in a rme-8 mutant. SNX-1 shows increased localization to endosomes in a rme-8 mutant and FRAP shows reduced recovery in rme-8 versus wild-type suggesting a role for RME-8 in removal of SNX-1. Finally, using Alphafold structural predictions, they find that the IWN domains appear to be hinges connecting alpha solenoid structures providing insight into how the missense mutations in the IWN3 and IWN4 could disrupt function and demonstrating that the N-terminus appears to have three PH like domains that explains why the IWN1/2 deletion would affect membrane localization.

Strengths

This is a novel study that provides mechanistic insight into how a widely conserved regulator of endosome trafficking functions and provides many new avenues for further explanation. This study is nice in that it employs a variety of techniques to dig into RME-8 function using phylogenetic analyses, genetics, localization studies, binding assays, and structural modeling. The claims are backed by the data.

Weaknesses

The paper was a little difficult to read given errors in figure callout, misnumbering of figures and figures being out of order, though these are minor. It would be interesting to known how the point mutations affect cargo recycling, but this is not particularly necessary for this study.

Below are a list of typos and errors that need to be corrected.

P13 No figure callouts for Fig S3 (the real S3)

S4 Figure is inserted between figures 4 and 5. Further, this is a supplement for figure 4, not figure 5.

S5 Figure is also labelled as S4.

P14 line 198 Figure S3 callout should be for Figure S4

P14 line 213 Figure 6E callout appears to refer to Figure 4G?

P16 line 268 S4 Figure callout should be S5

P16 line 269 The triple Lysine CRISPR mutant does not match that in S5 Figure legend

P16 line 270 The triple Lysine transgenic does match the residues listed in the Figure 5 legend.

P21 line 369, odd phrasing “to our analyze”

Figure 4C Label images as “assay” and “growth” as done in panels A and B

Figure 4D Only some of the color coding on amino acids in the alignment are described in the legend.

Figure 4F Describe for the color coding for the sequence.

Figure 4H Define “V” and “wt” in the figure legend. Label images as “assay” and “growth”. The assay and growth images appear to be reversed as compared to the other Y2H panels.

Figure 4 legend, line 4 Should 1955 be 1962?

Figure 4 legend, line 5 R mutant appears to grow despite claim that it does not.

Figure 5 legend, IWN4* triple mutant, where did this come from and isn’t 1966 an Asp and not a Lys?

Figure 5 legend Remove unnecessary P value indicators, there are no ** or **** in the figure. Same for other figure legends.

Figure 6G and H, inset described in the legend is missing

Figure 6I, no statistics

S3 Figure legend is incomplete

S5 Figure is labeled as a second S4

Reviewer #3: Summary

Cargos captured by early endosomes face two alternative fates. They are sorted either into the degradation pathway and deposited into the lysosomal lumen for degradation, or into the recycling pathways through recycling endosomes or Golgi. Previously, by studying early endosomes in the coelomocytes of the nematode C. elegans, these authors identified that the fate of the cargo is determined by whether it is at the proximity of one of the two adjacent and opposing microdomains established on the endosomal surface: the degradation microdomain coated by the ESCRT-0 complex, in which HRS/HGRS-1 is a major component, or the recycling microdomain coated by the RME-8/SNX-1 complex. RME-8 was proposed to further attract the Hsc70/DNAK protein (the uncoating protein) by the protein-protein interaction between the DNAJ domain of RME-8 and Hsc70. Each of these two microdomains occupies approximately half of the endosomal surface. They (Norris et al, 2017, PNAS) reported that the RME-8/SNX-1 complex acts to limit the spreading of the HRS microdomain. In snx-1(0) and rme-8(ts) mutants, increased endosomal coverage and intensity of HGRS-1–labeled microdomains, as well as increased total levels of HGRS-1 bound to membranes were observed. The HRS/HGRS-1 microdomain spreads to overlap with the recycling microdomain. On the other hand, in hgrs-1 loss-of-function mutants, the area covered by the RME-8/SNX-1 is not changed, indicating the active role of the recycling microdomain in establishing the microdomain pattern on endosomes.

In this new report, the authors have further investigated the mechanism of how RME-8 and SNX-1 establish the recycling microdomain and limit the spreading of the degradation microdomain. They analyzed the domain structure of RME-8 by alignment of the protein sequences of RME-8 family members in many species and by studying the AlphaFold prediction of the tertiary structure of RME-8. They further performed extensive structural-function analysis by employing three major approaches including (1) monitoring the endosomal localization pattern of the mutated or truncated proteins tagged with GFP-based reporters in various genetic backgrounds, (2) the yeast two-hybrid assay for protein-protein interaction, and (3) measuring the rescuing activity of the rme-8(b1023ts) mutant phenotype with multiple mutant forms of rme-8. These authors report that C-terminal to the DNAJ domain there are two IWN domains, IWN3 and 4. The region containing IWN3-IWN4 is able to bind to both the DNAJ domain of RME-8 and SNX-1. DNAJ and SNX-1 appear to compete with each other for the binding with IWN3-IWN4. In addition, mutations that change the charge of the amino acids in IWN3 and IWN4 both alter the binding affinity to DNAJ and SNX-1. Based on extensive mutational analyses, the authors propose a model of action. In this model, the IWN(3-4) region of one RME-8 molecule is proposed to interact with the DNAJ domain of a different RME-8 molecule, resulting in the formation of RME-8 oligomers, in which the DNAJ domain is hidden from interaction with Hsc70. By interacting with the IWN(3-4) domains, SNX-1 disrupts the inhibited conformation of RME-8, allowing the DNAJ domain to be exposed and accessible to Hsc70. The RME-8/Hsc70 complex then acts to uncoat the ESCRT-0 complex once it invades the RME-8 microdomain. Furthermore, this mode of action constitutes an activity cycle, since SNX-1 is also proposed to be a substrate of the RME-8/Hsc70 complex based on experimental evidence. The authors refer to this model as a “Ready Set Go model” for cyclical activation of RME-8.

I enjoyed reading this manuscript very much. How the fates of cargos are determined in the endocytic trafficking pathway was not well understood. This question is so universally important because if the regulatory mechanisms go awry, it would impact many developmental and health issues. The molecular mechanism proposed here is both original and of general interest. The model is well supported by experimental observations. The discovery will be welcomed by cell biologists, developmental biologists, and molecular biologists. I do not have any major concerns, yet there are a few specific issues that need to be further clarified. I would like to see my concerns addressed by either experiments (when specified) and/or additional explanations.

Specific concerns:

1. Lines 275-278, “Interestingly, the hyperactivity of RME-8(IWN3*) in reducing HGRS-1/Hrs levels on endosomes was blocked by endogenous wild-type RME-8, since we only observed this hyperactivity of RME-8(IWN3*) when endogenous RME-8 was removed by temperature shift of (b1023ts) mutants (Figure 5A” and B”, quantified in 5E)”.

This statement is given without showing the readers the proper control experiment, which is, what happens to the HGRS-1/Hrs levels in a wild-type strain expressing RME-8(IWN3*). This control is necessary.

2. Regarding the microdomain localization of RME-8(IWN3*), Figure 6 (D and J) indicates that RME-8(IWN3*) expressed in the wild-type strain occupies part of the endosomal surfaces. However, Figure 6 (F and L) indicates that when RME-8(IWN3*) is co-expressed with RME-8, the endosomal localization of RME-8(IWN3*) is further expanded, yet that of RME-8 remains limited in its microdomain. How to explain the expansion of RME-8(IWN3*)? This expansion cannot be explained by the proposed inhibitory effect of RME-8 towards RME-8(IWN3*) (Lines 282-285), as the area that RME-8 is localized to does not change and is smaller than that of RME-8(IWN3*).

3. The discovery that “SNX-1 may be a substrate for RME-8/Hsc70 assembly/disassembly activities, in addition to acting as an activator of RME-8 uncoating activity toward the degradative microdomain.” is novel and exciting. It supports the model of a cyclic regulation of the microdomain occupation of the RME-8/SNX-1 proposed by the authors. To strengthen this model, I wonder whether the three IWN4 mutations that increase the binding affinity between RME-8 and SNX-1 in the yeast two-hybrid assay would affect the microdomain localization of SNX-1 in worms. In particular, in the Discussion, it is stated that “IWN4* improves the interaction with SNX-1 and is not hyperactive hinting that release of SNX-1 maybe an important part of the activation cycle of

417 RME-8”.

Have these three mutations been tested for their in vivo effect on SNX-1 microdomain localization? If yes, what is the observation? If not, I would like to see them tested

4. What is the mechanism(s) that restricts RME-8 on part of the endosomes, in its own microdomain? Do the authors have any speculations? The authors hint in the Discussion that uncoating of SNX-1 by the DNAJ/Hsc70 complex might act to restrict the RME-8 microdomain. However, in snx-1(0) mutants, RME-8 localization is peripheral rather than expanding the entire endosomes. Is there an explanation for this phenotype?

5. Also on the issue regarding the mechanism that restricts RME-8 to only part of the endosomes, RME-8(�IWN(3+4)) expands to the entire endosome in the wild-type strain. Does this expansion depend on the function of the endogenous wild-type RME-8? To answer this question, which will help distinguish whether the (�IWN(3+4) deletion causes a total loss of RME-8 function, could the authors characterize a rme-8(�IWN(3+4))::GFP homozygous mutant (CRISPR-generated) or the rme-8(b1023ts) mutants expressing rme-8(�IWN(3+4))::GFP for the endosomal localization pattern?

6. There are two phenotypes that need to be distinguished from each other. In the 2017 PNAS paper, it was reported that in rme-8(b1023ts) mutants, HRS expands its microdomain to most part of the surface of an endosome. In this manuscript, it is reported that in rme-8(b1023ts) mutants, there is an HRS overaccumulation phenotype. Is the overaccumulation phenotype the same as the expansion of the HRS microdomain? The illustration shown in Figure 3J seems to indicate that these are two different phenotypes. The overaccumulation phenotype is illustrated as that HRS signal intensity is increased inside its microdomain without the expansion of the microdomain. Is my understanding correct? Furthermore, the expression of RME-8(�IWN(3+4)) in the wild-type strain is reported to cause the overaccumulation of HRS on its microdomain, without any sign of HRS-microdomain expansion. What is the mechanism of this HRS overaccumulation, if HRS is still restricted in its usual microdomain?

7. line 213, is Figure 6E the right figure to cite?

8. Lines 264-266,“Conversely, in vivo we observe SNX-1 colocalization with RME-8 is dramatically lower for RME-8(IWN3*) than with wild-type RME-8 (Figure 6F-G”, quantified in 6J, illustrated in 6K)”.

--should the figure citation be (Figure 6G-H, quantified in 6K, illustrated in 6J)?

9. Lines 268-270, “We tested CRISPR mediated endogenous alterations E1962K, and a triple lysine substitution at E1959, E1962, or N1966 (Figure S5)”.

-- Did not find Figure S5. Do you mean Figure S4?

-- There are two different figures that are labeled as Figure S4.

**Have all data underlying the figures and results presented in the manuscript been provided?**

Reviewer #1: None

Reviewer #2: Yes

Reviewer #3: Yes

PLOS authors have the option to publish the peer review history of their article (what does this mean?). If published, this will include your full peer review and any attached files.

Reviewer #1: No

Reviewer #2: No

Reviewer #3: No

---

## [Decision Letter · Decision Letter 1]

20 Sep 2022

Dear Dr Norris,

Thank you very much for submitting your Research Article entitled 'Mutagenesis and structural modeling implicate RME-8 IWN domains as conformational control points' to PLOS Genetics.

The manuscript was fully evaluated at the editorial level and by independent peer reviewers. The reviewers appreciated the attention to an important topic but identified some concerns that we ask you address in a revised manuscript. We therefore ask you to modify the manuscript according to the review recommendations. Your revisions should address the specific points made by each reviewer.

[LINK]

Yours sincerely,

Ken Sato

Guest Editor

PLOS Genetics

Gregory P. Copenhaver

Editor-in-Chief

PLOS Genetics

Dear Dr. Norris,

Manuscript (PGENETICS-D-22-00710R1), entitled "Mutagenesis and structural modeling implicate RME-8 IWN domains as conformational control points", which you submitted to PLOS Genetics, has been reviewed. The comments of the reviewer(s) are included at the bottom of this letter. The reviewers have recommended publication, but still suggest some minor revisions to your manuscript. Therefore, I invite you to respond to the reviewer 2' comments and revise your manuscript as much as possible.

Yours sincerely,

Ken Sato

Guest Editor

PLOS Genetics

Reviewer's Responses to Questions

**Comments to the Authors:**

Reviewer #1: Norris et al. have addressed the major criticisms and applied many of the reviewers suggestions to improve the presentation of the data. As before, it represents a significant and substantial advance in our understanding of RME-8.

Reviewer #2: My original comments on Norris et al were minor and mostly pertained to mistakes in the figures and the callouts etc. These have been corrected in the revised manuscript. However, the revised manuscript still has many errors. This is an excellent paper that is very difficult to read which if further exasperated by these errors. One addition that would enhance the paper would be a better description of how phenotypes are quantified as well as a supplementary file that includes the numbers behind the graphs.

Minor.

Fig 1B legend, indicate lipid binding domain (L), IWN domains (1-4) DNAJ (J). Obvious, but not always.

Fig S1B. Why are there numbers, lower case letters and spaces in the C. elegans RME-8 sequence? Seems like it would be more useful without these annotations from the multiple sequence alignment.

Fig 2A. Golgi are located in the cytoplasm as well as in the endosomes. Is this a stereotypical localization of Golgi in coelomocytes?

Line 532 What is the meaning of the double zeta?

Line 620 Platted? Do you mean plotted?

Fig 2. How is membrane to cytoplasm ratio measured? Do you draw a line across a vesicle membrane into the cytoplasm or draw a box in different regions of the coelomocyte? Do you measure one vesicle per animal? More details on how the differences are measured would be helpful for other doing similar experiments based on your work. Same with microdomain spread.

Since graphs are summaries of the data, an excel file with measurements for each of the graphs would increase transparency of the work.

The callout for Fig S2 is after the callout for Fig S3 making it a little confusing (I thought you were missing the callout till I found it later in the text).

Fig 3 and legend. Images A-C and D-F are labeled oppositely in the figure as they are described in the legend.

Fig 3N the arrow is overlapping with the text

Fig 4B legend D1657 mutant is not described.

Line 230. Should the Fig 4D callout be for 4C?

Fig 5F Text is overlapping with the drawing.

Line 295 Fig 5C callout should be for Fig S5C.

Fig 6D-F the inset of the higher magnification overlaps with D-F’’ which is visually unappealing. This should be physically separated or have a white outline that distinguishes it as separate from the underlying image.

Line 318 What is the “n” in the Fig 6 callout?

Are there callouts for Fig 6 J-L?

Line 333-334 Fig 7 callouts are incorrect, should be A,B and H

Reviewer #3: The authors have addressed all of my concerns with the revision and the response. Thank you for the great job!

**Have all data underlying the figures and results presented in the manuscript been provided?**

Reviewer #1: None

Reviewer #2: **No: **Numbers underlying the graphs and the stats should be included.

Reviewer #3: Yes

PLOS authors have the option to publish the peer review history of their article (what does this mean?). If published, this will include your full peer review and any attached files.

Reviewer #1: No

Reviewer #2: No

Reviewer #3: No

---

## [Editor Report · Decision Letter 2]

4 Oct 2022

Dear Dr Norris,

We are pleased to inform you that your manuscript entitled "Mutagenesis and structural modeling implicate RME-8 IWN domains as conformational control points" has been editorially accepted for publication in PLOS Genetics. Congratulations!

Yours sincerely,

Ken Sato

Guest Editor

PLOS Genetics

Gregory P. Copenhaver

Editor-in-Chief

PLOS Genetics

Comments from the reviewers (if applicable):

**Data Deposition**

http://datadryad.org/submit?journalID=pgenetics&manu=PGENETICS-D-22-00710R2

**Press Queries**

---

## [Editor Report · Acceptance letter]

20 Oct 2022

PGENETICS-D-22-00710R2 

Mutagenesis and structural modeling implicate RME-8 IWN domains as conformational control points 

Dear Dr Norris, 

We are pleased to inform you that your manuscript entitled "Mutagenesis and structural modeling implicate RME-8 IWN domains as conformational control points" has been formally accepted for publication in PLOS Genetics! Your manuscript is now with our production department and you will be notified of the publication date in due course.

With kind regards,

Zsuzsanna Gémesi

PLOS Genetics

On behalf of:
